# Factors associated with unfavorable treatment outcomes in patients with rifampicin-resistant tuberculosis in Colombia 2013–2015: A retrospective cohort study

Ninfa Marlen Chaves-Torres[1,2]☯*, Santiago Fadul[3‡], Jesus Patiño[1‡], Eduardo Netto[4☯]

1 Postgraduate Program in Medicine and Health, Federal University of Bahia, Salvador, Brazil, 2 Faculty of Medicine and Health Sciences, Military University Nueva Granada, Bogotá, Colombia, 3 Department of Communicable Diseases, Respiratory Diseases Team, National Institute of Health, Bogotá, Colombia, 4 Postgraduate Program in Medicine and Health, Federal University of Bahia, José Silveira Foundation, Salvador, Brazil

☯ These authors contributed equally to this work.
‡ These authors also contributed equally to this work.
* marlenchaveztdoc@yahoo.es

**Data Availability Statement:** All relevant data are within the manuscript and its Supporting Information files.

## Abstract

### Background

Multidrug- and rifampicin (RMP)-resistant tuberculosis (MDR/RR-TB) requires prolonged and expensive treatment, which is difficult to sustain in the Colombian health system. This requires the joint action of different providers to provide timely health services to people with TB. Identifying factors associated with unfavorable treatment outcomes in patients with MDR/RR-TB who received drug therapy between 2013 and 2015 in Colombia can help guide the strengthening of the national TB control program.

### Method

A retrospective cohort study was conducted with all patients who received treatment for MDR/RR-TB between January 2013 and December 2015 in Colombia who were registered and followed up by the national TB control program. A multivariate logistic regression model was used to estimate the associations between the exposure variables with the response variable (treatment outcome).

### Results

A total of 511 patients with MDR/RR-TB were registered and followed up by the national TB control program in Colombia, of whom 16 (3.1%) had extensive drug resistance, 364 (71.2%) had multidrug resistance, and 131 (25.6%) had RMP monoresistance. The mean age was 39.9 years (95% confidence interval (CI): 38.5–41.3), most patients were male 285 (64.6%), and 299 (67.8%) were eligible for subsidized health services. The rate of unfavorable treatment outcomes in the RR-TB cohort was 50.1%, with rates of 85.7% for patients with extensive drug resistance, 47.6% for patients with multidrug resistance, and 52.6% for patients with RMP monoresistance. The 511 MDR/RR-TB patients were included in

**Funding:** This study was financed in part by the "Coordenação de Aperfeiçoamento de Pessoal de Nível Superior – Brasil (CAPES)" – Finance Code 001. There was no additional external funding received for this study.

**Competing interests:** The authors have declared that no competing interests exist.

bivariate and multivariate analyses, patients age ≥ 60 years (crude odds ratio (ORc) = 2.4, 95% CI 1.1–5.8; adjusted odds ratio (ORa) = 2.7, 95% CI 1.1–6.8) and subsidized health regime affiliation (ORc = 3.6, 95% CI 2.3–5.6; ORa = 3.4, 95% CI 2.0–6.0) were associated with unfavorable treatment outcomes.

## Conclusion

More than 50% of the patients with MDR/RR-TB in Colombia experienced unfavorable treatment outcomes. The patients who were eligible for subsidized care were more likely to experience unfavorable treatment outcomes. Those who were older than 60 years were also more likely to experience unfavorable treatment outcomes.

## Introduction

Multidrug-resistant tuberculosis (MDR-TB) is defined as TB resistant to at least rifampicin (RMP) and isoniazid (INH), and extensively drug-resistant TB (XDR-TB) is defined as MDR-TB plus resistance to at least one quinolone and to a second-line injectable drug used to treat TB (capreomycin, kanamycin, or amikacin) [1]. MDR/RR-TB poses a threat to TB control–patients have worse long term prognosis, and treatment is more likely to be expensive and present toxicity. Lack of treatment support (including DOT) and limited treatment adherence have both been associated with the development of MDR/RR-TB. Other contributing factors include social and cultural barriers to access, such as living in rural areas and the absence of private health insurance The World Health Organization (WHO) estimated 484,000 cases (range, 417,000–556,000) of MDR/RMP-resistant (RR)-TB globally in 2018 compared to 160,684 cases in 2017, with 156,071 patients undergoing treatment in 2018 compared to 139,114 in 2017. Although more people underwent treatment in 2018, they accounted for only 32% of the estimated incident cases, and the treatment success rates were 56% for MDR/RR-TB and 39% for XDR-TB [2].

In 2017, Colombia had approximately 16,000 estimated cases of TB and thus ranked fifth among the countries with the largest numbers of TB cases and sixth among the countries with the highest estimated burden of MDR/RR-TB in the Americas [3]. The proportion of patients experiencing unfavourable treatment outcome is higher in MDR/RR-TB than in TB with drug sensitive bacilli. Other factors, including education, race, age, drug use, history of second-line treatment, resistance to fluoroquinolones, positive sputum smear after two months of treatment, and XDR-TB are associated with higher likelihood of unfavourable outcome [4–6], and coinfection with HIV has been specifically associated with death [5].

Studies on the treatment outcomes for MDR/RR-TB in Colombia are scarce and not representative of the Colombian population. MDR/RR-TB often necessitates prolonged and more expensive treatment, which is challenging to sustain in the current Colombian health care system. Identifying factors associated with unfavourable treatment outcome for patients with MDR/RR-TB in Colombia, may support design and delivery of the national TB control programs. The objective of this study was to identify factors associated with unfavorable treatment outcomes in patients with MDR/RR-TB who began treatment between 2013 and 2015 in Colombia.

## Methods

### Study site

Colombia is an upper-middle income country (Human Development Index 0.761), however there remains significant wealth inequality [7]. In this unequal country, the public health

system encounters difficulties in responding to the economic and social demands involved in the treatment of MDR/RR-TB.

The General Health and Social Security System (Sistema General de Seguridad Social en Salud–SGSSS) in Colombia covers approximately 95% of the population [8]. This organization separates insurance and administration of financial resources, from the direct service provision management of its members [9]. There are two affiliation regimes—a *contributive regime* that covers 47% of affiliates, including all employees and independent persons with the ability to pay who make a monthly contribution to health and retirement, and a *subsidized regime*, which covers the remaining 53% of the population [10], primarily including poor and vulnerable people who rely on public healthcare services [11]. However, people who qualifying for subsidized health care services are not necessarily beneficiaries of social protection programs. This model includes three entities: the *state (government)*, which is responsible for coordination, direction, and control; *Private health insurance companies*, which are private companies that manage the population's resources and act as intermediaries, organizing and guaranteeing the provision of health services; and *health service providers*, including hospitals, and community clinics', that provide health services directly to users [12, 13].

The Colombian Government's National Public Health Plan priorities national health promotion and disease prevention, including infectious diseases such as TB, Leprosy and Malaria. Therefore, TB is a disease of public health interest, and its diagnosis, treatment and follow-up are publicly funded as a result [14].

## Study type and population

A retrospective cohort study was conducted with all patients who were notified and diagnosed with MDR/RR-TB and receiving treatment between January 1st, 2013, and December 31st, 2015, in Colombia. Data for treatment outcomes were collected until December 2017, assuming a period of at least two years bearing in mind that in these cases the total treatment duration should be at least 20 months. Patients were excluded if no record of treatment results up to this date was available. The Ministry of Health and Social Protection of Colombia (MINSALUD) provided the participants' data in a database with anonymised information.

## Definitions of terms

In Colombia, suspected drug resistance is defined by the presence of the following risk factors: (a) treatment failure; (b) readmission with positive bacteriology after having met criteria for treatment abandonment; and (c) early relapse and exposure (contact) to a person with confirmed MDR/XDR-TB. In these cases, culture and a drug sensitivity test (DST) are requested. MDR-TB is defined as TB resistant to isoniazid and rifampicin with or without resistance to other first-line drugs. The diagnosis of drug-resistant TB is established only by confirmation of resistance in vitro or by molecular tests to one or more anti-TB drugs. If the therapeutic decision upon suspicion and/or after confirmation leads to the prescription of a treatment scheme different from the standard treatment for drug-sensitive TB, the case is reported to the national program as drug-resistant TB [15].

DSTs were performed in the national network of laboratories using the Löwenstein-Jensen method, the Bactec MGIT 960® instrument, molecular screening by the Genotype® MDR-plus method, the Xpert TB RIF® test, and DNA detection tests based on line probe analysis according to the Report of Activities Performed by the National Network of Laboratories for the surveillance of *Mycobacterium tuberculosis* resistance to anti-TB drugs [16, 17]. Resistance to anti-TB drugs was defined according to the MINSALUD circular 007/2015: XDR was defined as resistance to RMP, INH, a fluoroquinolone, and at least one second-line injectable

drug; MDR was defined as simultaneous resistance to at least INH and RMP; polyresistance was defined as resistance to more than one drug without including simultaneous resistance to INH and RMP; monoresistance was defined as resistance to a single anti-TB drug; and RMP resistance was defined as any resistance to RMP, including monoresistance, polyresistance, multidrug resistance, or extensive drug resistance [18].

Treatment outcome was considered a dichotomous variable (unfavorable outcome). Treatment was favorable when the patient was considered cured or finished treatment. An unfavorable outcome was considered when the result was lost in follow-up, treatment failure, or death. The guidelines for the tuberculosis and leprosy management program in Colombia provide the following definitions: **Cured**–treatment completion without evidence of failure and three or more consecutive negative cultures at an interval of at least 30 days after the intensive phase; **Finished Treatment**–cured but without evidence of three or more negative cultures; **Treatment Failure**–treatment suspension, the need for a permanent scheme change or a change in at least two anti-TB drugs due to lack of conversion at the end of the intensive phase, bacteriological reversal in the continuation phase after conversion to a negative status, evidence of acquired additional resistance to fluoroquinolones or second-line injectable drugs, or adverse drug reactions; **Death**–death for any reason during the course of treatment; **Lost in follow-up**–treatment interruption for two consecutive months or more; and **Not Evaluated**–no treatment outcome assigned (including cases "transferred" to another treatment unit, cases with unknown treatment outcomes, and patients who were still receiving treatment as of December 2017, which resulted in an unknown final outcome) [18].

## Statistical analysis

The supplied database was imported into the statistical program SPSS 18.0® for statistical analysis. A descriptive analysis was performed, and proportions were compared using the chi-square test (nominal variables). A multivariate logistic regression model was used to analyze associations between the exposure variables (age, sex, ethnicity, the site of TB, health regime, treating HCI, level of care, the method used for diagnosis, and the type of resistance) with the response variable (treatment outcome). Variables with p-values < 0.05 in the bivariate analysis were included in the multivariate analysis. We did not include or exclude a priori variables from the multivariate analysis.

## Ethical considerations

This study was conducted in accordance with the principles expressed in the Declaration of Helsinki and following the technical, administrative, and scientific standards for health research in Colombia approved by the Ministry of Social Protection under Resolution no. 008430 of October 1993 and is part of the research project "Treatment cohort of patients with tuberculosis in Brazil, Colombia and Peru," which was approved by the Ethics Committee of Climério de Oliveira Maternity, Federal University of Bahia (CAAE 74672317.4.0000.5543).

## Results

### Descriptive analysis

A total of 41,144 patients were diagnosed with TB in Colombia between 2013 and 2015, 950 (2.3%) of whom were diagnosed with drug resistance, and among these patients, 511 (53.8%) were diagnosed with RR-TB, 16 (3.1%) of whom exhibited extensive drug resistance, 364 (71.2%) exhibited multidrug resistance, and 131 (25.6%) exhibited monoresistance to RMP; 167 cases were diagnosed in 2013, 181 in 2014, and 163 in 2015. A total of 439 patients did not

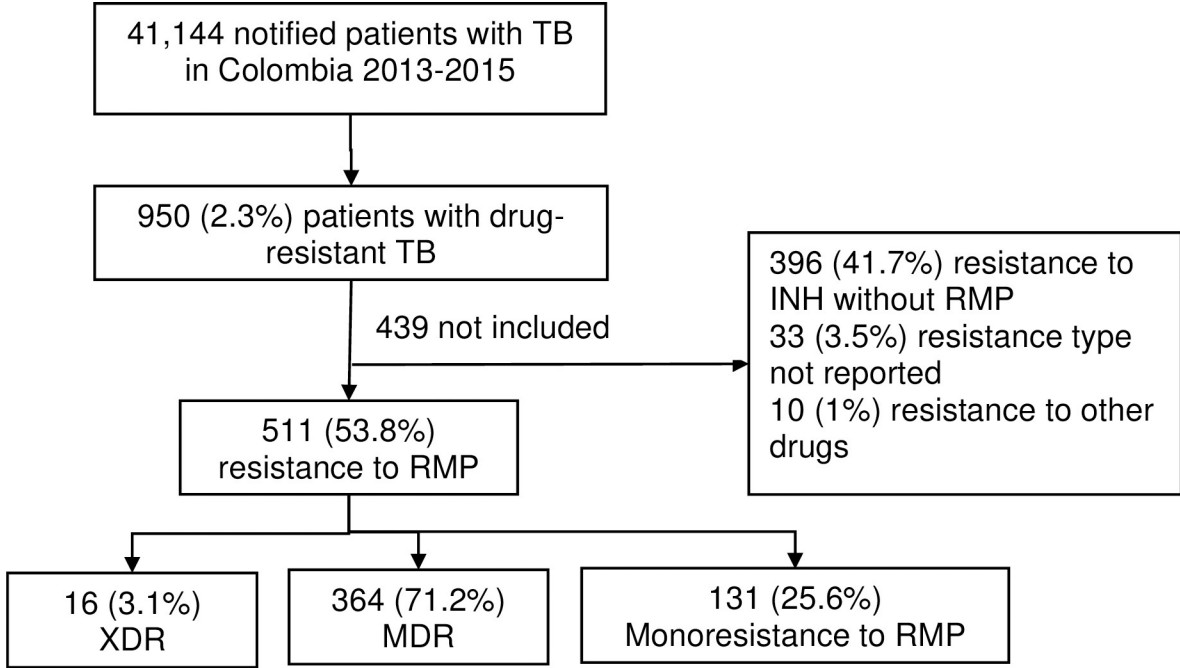

**Fig 1. Flowchart of the selected study population with MDR/RR-TB in Colombia, 2013–2015.** (TB: tuberculosis; XDR: extensive drug resistance; MDR: multidrug resistance; RMP: rifampicin; INH: isoniazid; INH without RMP: mono- or polyresistance to isoniazid that does not include rifampicin; other resistance: resistance to any antituberculosis drug other than RMP and INH).

meet the inclusion criteria in this cohort (Fig 1). The characteristics of the excluded patients were not different from those of the patients included in the analysis (S1 Table).

Table 1 shows the sociodemographic, clinical, and laboratory characteristics as well as the treatment outcomes of patients with RR-TB in Colombia. The mean age was 39.9 years (95% confidence interval (CI): 38.5–41.3), most patients were male (n = 285; 64.6%), 299 (67.8%) patients were affiliated with the subsidized health regime, 309 (70.1%) patients received treatment at a primary level HCI, and 275 (63.7%) patients were treated in public HCIs. A total of 64% of the cases were reported in 10 major cities, which are also the most populous cities in the country; Medellín accounted for 122 (23.9%) of all cases.

## Treatment outcomes and factors associated with unfavorable outcomes

Overall, 221 (50.1%) patients with RR-TB showed unfavorable treatment outcomes. 122 (27.7%) of whom lost to follow up, 12 (2.7%) had treatment failure, 87 (19.7) died during treatment, and 220 (49.9%) had treatment success. Patients with XDR-TB had a higher rate of unfavorable outcomes than patients with MDR-TB and monoresistance to RMP at 85.7% vs 47.6% and 52.6%, respectively (p = 0.018). (Table 1).

Age $\geq$ 60 years showed association with unfavorable outcomes (crude odds ratio (ORc) = 2.4, 95% CI 1.1–5.8; adjusted odds ratio (ORa) = 2.7, 95% CI 1.1–6.8). Male sex was also associated with unfavorable outcomes in the bivariate analysis, but in the multivariate analysis, this association was not significant (ORc = 1.5, 95% CI 1.1–2.2; ORa = 1.2, 95% CI, 0.8–1.9) (Table 1).

Patients affiliated with the subsidized regime were more likely to have unfavorable treatment outcomes (ORc = 3.6, 95% CI 2.3–5.6; ORa = 3.4, 95% CI 2.0–6.0). Patients treated in public HCIs were also more likely to have unfavorable treatment outcomes, but this association was not maintained in the multivariate analysis (ORc = 2.4, 95% CI 1.6–3.7; ORa = 1.2,

**Table 1. Sociodemographic, clinical, and laboratory characteristics and treatment outcomes of patients in the MDR/RR-TB cohort in Colombia, 2013–2015.**

| | Favorable n = | | Unfavorable n = | | p[a] | Bivariate A | | Multivariate A | |
|---|---|---|---|---|---|---|---|---|---|
| | **220** | **%** | **221** | **%** | | **ORc[b]** | **95% IC** | **ORa[c]** | **95% IC** |
| **Age (years)** | | | | | | | | | |
| < 20 | 24 | (61.5) | 15 | (38.5) | | Ref[d] | | Ref | |
| 20 to 39 | 95 | (51.1) | 91 | (48.9) | 0.236 | 1.5 | (0.8–3.1) | 1.8 | (0.8–3.8) |
| 40 to 59 | 82 | (48.8) | 86 | (51.2) | 0.154 | 1.6 | (0.8–3.4) | 1.6 | (0.7–3.5) |
| ≥60 | 19 | (39.6) | 29 | (60.4) | 0.043 | 2.4 | (1.1–5.8) | 2.7 | (1.1–6.8) |
| **Sex** | | | | | | | | | |
| Female | 88 | (65.4) | 68 | (43.6) | | Ref | | Ref | |
| Male | 132 | (46.3) | 153 | (53.7) | 0.04 | 1.5 | (1.1–2.2) | 1.2 | (0.8–1.9) |
| **Self-reported ethnicity** | | | | | | | | | |
| Ethnicity, not referred | 171 | (47.9) | 186 | (52.1) | | Ref | | | |
| Indigenous | 6 | (50.0) | 6 | (50.0) | 0.886 | 0.9 | (0.3–2.9) | | |
| Afro-Colombian | 43 | (59.7) | 29 | (40.3) | 0.069 | 0.6 | (0.4–1.0) | | |
| **Form of tuberculosis** | | | | | | | | | |
| Extrapulmonary | 19 | (55.9) | 15 | (44.1) | | Ref | | | |
| Pulmonary | 201 | (49.4) | 206 | (50.6) | 0.468 | 0.8 | (0.4–1.6) | | |
| **Health regime** | | | | | | | | | |
| Contributive | 100 | (70.4) | 42 | (29.6) | | Ref | | Ref | |
| Subsidized | 120 | (40.1) | 179 | (59.9) | <0.001 | 3.6 | (2.3–5.6) | 3.4 | (2.0–6.0) |
| **Treating HCI[e]** | | | | | | | | | |
| Private | 101 | (64.3) | 56 | (35.7) | | Ref | | Ref | |
| Public | 117 | (42.5) | 158 | (57.5) | <0.001 | 2.4 | (1.6–3.7) | 1.2 | (0.7–2.0) |
| **Level of care** | | | | | | | | | |
| Primary | 165 | (53.4) | 144 | (46.6) | | Ref | | | |
| Secondary and tertiary | 47 | (41.6) | 66 | (58.4) | 0.03 | 1.6 | (1.0–2.5) | | |
| **Method used to diagnose resistance** | | | | | | | | | |
| L.J[f] Proportions | 20 | (44.4) | 25 | (55.6) | | Ref | | | |
| Bactec MGIT[g] | 97 | (50.0) | 97 | (50.0) | 0.502 | 0.8 | (0.4–1.5) | | |
| Lipa[h] | 63 | (52.5) | 57 | (47.5) | 0.358 | 0.7 | (0.3–1.4) | | |
| Real-time PCR[i] | 40 | (48.8) | 42 | (51.2) | 0.640 | 0.8 | (0.4–1.7) | | |
| **Type of resistance** | | | | | | | | | |
| Monoresistance RMP [j] | 54 | (47.4) | 60 | (52.6) | | Ref | | Ref | |
| XDR[k] | 2 | (14.3) | 12 | (85.7) | 0.032 | 4.2 | (0.8–20.6) | 4.5 | (0.9–22.6) |
| MDR[l] | 164 | (52.4) | 149 | (47.6) | 0.358 | 0.7 | (0.4–1.1) | 0.7 | (0.4–1.1) |

[a]p-value calculated by the chi-square test for favorable vs. unfavorable outcomes

[b]crude odds ratio

[c]adjusted odds ratio

[d] reference

[e]health care institution

[f]Löwenstein-Jensen

[g]mycobacterial growth indicator

[h]line probe assay

[i]polymerase chain reaction

[j]rifampicin

[k]extensive drug resistance

[l] multidrug resistance. The percentage reported is that of the rows. In the logistic regression model, the following variables were entered in a single step: health regime affiliation, type of resistance to rifampicin, treating HCI, level of care, age, and sex

95% CI 0.7–2.0). In addition, receiving treatment in secondary and tertiary level HCIs was associated with unfavorable results, although the association was not significant (ORc = 1.6; 95% CI: 1.0–2.5) (Table 1).

## Discussion

An overall therapeutic success rate of 49.9% was found in this MDR/RR-TB cohort. 47.4% among patients with monoresistance to RMP, 52.4% among patients with MDR-TB, and 18.6% among patients with XDR-TB. These success rates are lower than those reported worldwide in 2015, with success rates of 55% for cases of MDR-TB and 35% for XDR-TB cases [1]. In a recent meta-analysis of 74 studies including 17,494 participants, the treatment success rates were 26% for patients with XDR-TB and 60% for patients with MDR-TB [19]. Another meta-analysis including studies from 25 countries reported a success rate of 61% for MDR-TB [20]. Studies conducted in Peru and Brazil also found success rates of 60% and 58.1% for MDR-TB, respectively, and 18.6% for XDR-TB [5, 21, 22]. These values are within the average range for countries in the Americas, where the treatment success rate for MDR/RR-TB was 56% in a 2015 cohort [3].

The association between the health regime and the TB treatment outcomes in Colombia is relevant. Individuals affiliated with the government-subsidized health regime, people with low economic resources without the ability to pay contributions to the health system, were more likely to have unfavorable treatment results (ORc = 3.6, 95% CI 2.3–5.6; ORa = 3.4, 95% CI 2.0–6.0). Although the diagnosis and treatment of MDR/RR-TB in Colombia are free for all patients regardless of their health regime affiliation, discrepancies remain in the outcomes of treatment. We speculate that in addition to the social and economic deprivation to which the population of the subsidized regime may be exposed, there are differences in the care provided to patients between the two health affiliation regimes. Which can be considered an organizational barrier, a factor hindering initial contact with health services (entry barriers) and timely care after a patient enters the health center (interior barriers). Inequalities related to health regime affiliations have been identified previously a national level; for patients receiving care in the subsidised healthcare system, higher under-5 mortality, maternal mortality and mortality related to communicable disease has already been observed [14, 23].

In addition, patients incur out-of-pocket expenses, which may be direct, such as those related to transportation and examinations or consultations or indirect, such as an inability to work due to the disease [24]. These expenses in in low-income families (As is the case for families in the subsidized regime) can create significant barriers to both access to the TB control program and adherence with TB treatment. A study carried out in Bogotá, the capital of Colombia, showed that a lower household income level corresponded to a greater likelihood of incurring catastrophic health costs [25].

In Colombia, qualifying to be a beneficiary of the subsidized health regime is synonymous with belonging to the poorest and most vulnerable population, and TB has a direct relationship with poverty and social exclusion. This relationship between poverty and TB was studied in Brazil, where the authors concluded that TB patients who were beneficiaries of a government cash grant, in this case the Bolsa Familia Program (PBF), were more likely to experience favourable outcome in the TB treatment [26, 27].

In Colombia, in capital cities such as Manizales, Pereira, and Armenia, most patients with TB have been reported to belong to the subsidized regime [28]. Likewise, in Medellín, patients who experienced successful TB treatment were found to be mostly affiliated with the contributive regime [29] according to our results. In Bogota in 2001, the association between the health regime and lost to follow up of TB treatment was also described. Cases of lost to follow up were more frequent among patients affiliated with the subsidized regime [30]. Other factors

associated with unfavorable TB treatment outcomes have also been studied, such as a lack of treatment adherence, the duration of therapy, and the patient's follow-up time in the tuberculosis control program [9, 28]. However, this study is the first to document factors associated with unfavorable outcomes of MDR/RR-TB treatment in Colombia.

Receiving care and treatment in public HCIs was associated with unfavorable outcomes (ORc = 2.4, 95% CI 1.6–3.7; ORa = 1.2, 95% CI 0.7–2.0). This association may be related to an affiliation with the subsidized regime because in Colombia, individuals affiliated with the subsidized regime are served mostly in public HCIs [9]. Care in secondary and tertiary level health facilities was associated with unfavorable outcomes (ORc = 1.6, 95% CI 1.0–2.5), possibly because primary level HCIs provide care to outpatients who have a lower risk of complications and death than patients treated in secondary and tertiary level hospitals.

Age and sex have been previously studied as factors associated with unfavorable TB treatment outcomes. In the present cohort, age greater than 60 years or older (ORc = 2.4, 95% CI 1.1–5.8; ORa = 2.7, 95% CI 1.1–6.8) and male sex (ORc = 1.5, 95% CI 1.1–2.2; Ora = 1.2, 95% CI 0.8–1.9) were associated with unfavorable treatment outcomes for MDR/RR-TB. This association has been previously described for TB in Colombia [9, 28, 31] and for MDR-TB in Brazil [22].

This study was carried out with secondary data (the PNCTB database) obtained from reliable PNCTB records but with a limited number of variables. A multivariate analysis was performed to highlight the factors that affecting tuberculosis treatment outcomes. However, one limitation was the impossibility of studying other factors that may influence the outcomes of treatment for MDR/RR-TB, such as coinfection with HIV, comorbidities such as diabetes, habits such as alcohol, tobacco, and psychoactive substance use, and socioeconomic factors such as family income, as these variables were not considered in the data provided. The absence of these variables may lead to overestimation or underestimation of the TB treatment outcomes in this cohort of patients. Due to a selection bias, which prevents extrapolating our results to the Colombian population with any of the aforementioned conditions.

The missing data were 70, representing a loss of 13.7%, which is acceptable, and the missing cases did not have different characteristics compared to those included in the analysis. However, the results obtained are likely representative of the Colombian population with MDR/RR-TB because all cases treated in the country over a period of three years were analyzed.

Managing public health databases remains a challenge for many countries. Although Colombia already has a detailed database for patients with drug-resistant TB, it can still be improved the quality of the recorded data to reduce lost data and facilitate collection of the largest number of variables possible in reports for researchers who request these data.

In conclusion, in Colombia, 50.1% of patients with RR-TB who started treatment between January 2013 and December 2015 showed unfavorable treatment outcomes, and 19.7% died during treatment. Additionally, 85.7% of patients with XDR-TB and 47.6% of patients with MDR-TB had unfavorable results. Patients who were eligible for subsidized care or were older than 60 years were more likely to experience unfavorable treatment outcomes.

Our results suggest that in addition to the conditions of vulnerability to which the Colombian population affiliated to the subsidized health care system may be exposed, the structure of the Colombian health care model may influence MDR-TB treatment outcomes, which should be considered in the design of strategies to reduce burden of MDR/RR-TB.

## Supporting information

**S1 Table. Sociodemographic, clinical, and laboratory characteristics of included and excluded patients in the MDR/RR-TB cohort in Colombia, 2013–2015.**
(DOCX)

## Acknowledgments

We thank the Ministry of Health and Social Protection and the National Program for the Control of Tuberculosis in Colombia for providing the data analyzed in this study.

## Author Contributions

**Conceptualization:** Ninfa Marlen Chaves-Torres, Eduardo Netto.

**Formal analysis:** Ninfa Marlen Chaves-Torres, Eduardo Netto.

**Investigation:** Ninfa Marlen Chaves-Torres, Eduardo Netto.

**Methodology:** Ninfa Marlen Chaves-Torres, Eduardo Netto.

**Writing – original draft:** Ninfa Marlen Chaves-Torres, Eduardo Netto.

**Writing – review & editing:** Ninfa Marlen Chaves-Torres, Santiago Fadul, Jesus Patiño, Eduardo Netto.

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
