## [Decision Letter · Decision Letter 0]

17 Aug 2020

PONE-D-20-18648

Factors associated with unfavorable treatment outcomes in patients with rifampicin-resistant tuberculosis in Colombia

PLOS ONE

Dear Dr. Chaves Torres,

Thank you for submitting your manuscript to PLOS ONE. After careful consideration, we feel that it has merit but does not fully meet PLOS ONE’s publication criteria as it currently stands. Therefore, we invite you to submit a revised version of the manuscript that addresses the points raised during the review process.

While the research is an excellent use of programmatic data and is in an important area in the field of TB, there are a number of areas in which it could be improved for re-review and further consideration of publication.

Please can you systematically address the reviewers' comments paying special attention to:

1) Enhancing the literature review and introduction with relation to social determinants of TB and their further association with adverse TB treatment outcomes

2) Clarify some parts of the methods including: inclusion and exclusion criteria; evidence informing inclusion of independent variables into your regression model (e.g. method of diagnosis); and methods used to arrive at your adjusted model

3) Improvements to the written text and flow

4) Adjustments to the Tables and Figures (see Reviewer 2's comments) to ensure that only the most relevant and applicable information is included and the tables and figures are supportive of the main study objectives.

We look forward to receiving your revised manuscript.

Kind regards,

Tom E. Wingfield

Academic Editor

PLOS ONE

Reviewers' comments:

Reviewer's Responses to Questions

**Comments to the Author**

1. Is the manuscript technically sound, and do the data support the conclusions?

Reviewer #1: Partly

Reviewer #2: Yes

2. Has the statistical analysis been performed appropriately and rigorously? 

Reviewer #1: Yes

Reviewer #2: Yes

3. Have the authors made all data underlying the findings in their manuscript fully available?

Reviewer #1: Yes

Reviewer #2: Yes

4. Is the manuscript presented in an intelligible fashion and written in standard English?

Reviewer #1: No

Reviewer #2: No

5. Review Comments to the Author

Reviewer #1: See attached file:

Major Revisions

- Statistical Analysis in Methodology

Please mention clearly that you first performed bivariate analysis, and whether thereafter included variables with p value < 0.05

Please clarify whether, and which, variables were included or excluded a priori from the multivariate analysis to clarify the model building process

Please specify which confounders were included in the analysis here, and which (if any) registry variables were excluded

- Descriptive Analysis

Were any other variables excluded due to missing data? Would be helpful to specify % of missing data here

Figure 1: Can you clarify why you decided to exclude 396 individuals with INH resistant TB? Did you perform a separate analysis which included this group to see how outcomes differed?

Table 1: How and why did you choose to disaggregate age into <20, 20 to 39, 40 to 59 and over 60 – it appears the majority are in the 20-59 bracket and valuable information may be lost by grouping into such large categories. Did you perform a sensitivity analysis using smaller age intervals?

There are only 16 patients with XDR TB; are you convinced this sample is large enough to make meaningful conclusions about the rate of unfavourable treatment outcomes in this group?

Please rationalise why you have included method of diagnosis in your analysis, how do you anticipate this would be associated with treatment outcome?

- Results: Treatment outcomes and factors associated with Unfavourable outcomes

Consider using primary healthcare as reference group in this analysis; it seems unsurprising here that primary healthcare is a ‘protective factor’ as surely this variable is just a proxy for patients being ‘less sick’. I would be more interested to see whether those receiving secondary/tertiary level care had better or worse outcomes – does being in a tertiary centre result in more specialised care or are patients sicker?

- Discussion

‘The strong association between health regime and TB treatment outcomes in Colombia is relevant, as individuals affiliated with the government-subsidized health regime, i.e., the poorest and most vulnerable population, had a higher probability of presenting unfavourable treatments’ and ‘The conditions of poverty and vulnerability of the population affiliated with the subsidized regime may be associated with different barriers accessing basic and health services’ - This feels like a leap; given your model does not include household income, education level, living conditions – isn’t qualifying for subsidised healthcare a proxy for socioeconomic deprivation in this analysis? It may be difficult to confirm whether the discrepancy in treatment outcome for this group relates to barriers to access, quality of care, or other social factors for unfavourable outcome. Consider rephrasing and discussing these limitations more fully.

‘Therefore we carry out a careful adjustment of possible confounding factors, seeking to reduce biases’- what kind of bias did you seek to reduce, and what other confounders should have been included? Please explain how your results may have been affected (e.g. over/underestimation of odds ratios) by these confounders being omitted

I think you need to address other limitations to this study e.g. limited sample size, lack of household level characteristics, lack of information on the preceding treatment regimen, and other confounding variables e.g. markers of socioeconomic deprivation and education which are not recorded in this registry, and limited information provided about ethnicity and race. Might be helpful to consider who is not captured by this registry

- Conclusions

Try to align the conclusions more closely with your results and discussion

Minor Revisions

- Title

Could be tweaked to correspond directly to its contents, there is some interchangeable reference between MDR-TB, DR-TB, XDR-TB, RR-TB and RMP. Would also be good to specify that it is a retrospective cohort study and registry based in the title

- Abstract

Background: could you clarify what you mean by ‘exerts great pressure on the complex Colombian health system’, e.g. interesting to know whether you are referencing treatment costs, lengthy treatment, inadequate infrastructure?

Methodology: If able please specify the study inclusion criteria, and that the study uses registry level data in the abstract. Are you including those who have received previous treatment?

Results: Language used in the results is inconsistent with the methodology in the abstract, would be helpful to review the sentences ‘511 patients who started treatment for MDR/RR-TB’ and ‘A total of 511 patients were diagnosed with RR-TB in Colombia’. Suggest reviewing the language ‘affiliated with the subsidized health regime’, for example could state ‘individuals who qualified for subsidised health care services’

Consider making it clearer in the abstract whether you included all patients with DR-TB in your regression model, or whether you performed sub-group analysis comparing MDR-TB, RMP and XDR-TB

Suggest reviewing key words to ensure they relate more closely to your paper, e.g. consider dropping ‘associated factors’

- Introduction

Suggest rephrasing ‘these conditions represent public health problems around the world’, and relating more directly to the challenge that DR-TB poses

Line 61 – are these definitely marked ‘improvements’?

Suggest potential restructuring, and position paragraph 3 first – strengthen argument as to why this is a particular public health concern in Colombia

Please ensure that in-text citations are inside of the sentence, e.g. before full stop.

‘In 2018, 205 67 cases were notified out of an estimated 580, for a detection rate of 35.3%, similar to 68 the detection rate of 32% reported worldwide’ – please clarify which cases you are referring to; it sounds like you suspect there is underreporting of DR-TB in Colombia?

Several references to ‘pressure’ and the ‘complex Colombian health System’ – please clarify in which respect, e.g. are you referencing lack of resources, infrastructure, workforce?

Suggest maintaining consistent language, e.g. stick with treatment, not mentioning pharmacotherapy in the study objective

- Methods

o Study Site

You may be able to better summarise or condense the provision of healthcare in Colombia by using a figure or schematic

Please confirm whether subsidised healthcare includes additional support, e.g. social protection measures

Suggest paragraph 1 e.g. line 89-93 could be better summarised and more clearly convey the importance of understanding DR-TB in this study setting

Could be helpful to mention earlier that the direct costs of treatment are ‘free’ at point of care, but there are hidden costs

Also please confirm that TB is a notifiable disease in Colombia

o Study Type and Population

Please consider specifying whether the dataset was anonymised or de-identified

Helpful to provide exact dates of study time frame, given you later state it might have been too early for treatment outcomes to be recorded for those where its treatment outcome was missing

Please confirm when treatment outcome was recorded (e.g. time to follow-up), and how

Notification and diagnosis are used interchangeably, helpful to stick to one term only as these can mean slightly different things

o Statistical Analysis in Methodology

Reconsider the language ‘analyze the interaction’, this does not really reflect the statistical analysis reported in the results – unless you did examine interaction terms? Otherwise suggest ‘to estimate the association between…’

Please specify which confounders were included in the analysis here, and which (if any) registry variables were excluded

o Definition of Terms

Suggest restructuring to first define outcome variables e.g. ‘favourable’ and ‘unfavourable’ and thereafter how DR-TB is defined in Colombia. It is slightly misleading to first define the clinical rationale needed to ‘suspect’ DR-TB, followed by criteria for the diagnosis of DR-TB – as not totally clear on how DR-TB was defined in your study inclusion criteria

Specify what you mean by the ‘Colombian National Policy’

- Results

o Descriptive Analysis

Figure 2: Is it necessary to show rates of both favourable and unfavourable treatment outcomes on the same graph? Please include y axis for scale. Consider excluding this figure, I am not sure that it helps to answer the study objective.

What is the derivation of ‘afro-descendant’? Perhaps consider changing to Afro-Colombian. Who comprises the ‘other’ group – it seems this is the ‘majority’ – How has ethnicity been defined here?

Reconsider phrasing ‘first’, ‘second’ and ‘third’ care as ‘primary’, ‘secondary’ and ‘tertiary’

On line 211, please reconsider ‘were affiliated with the subsidized social security and health regime’. Does the subsidised health care also include social protection and other welfare benefits? This contradicts the earlier definition

o Treatment outcomes and factors associated with Unfavourable outcomes

Describing ‘the treatment success rate with respect to age showed an inversely proportional trend’ seems like a bit of a leap; could be worth rephrasing, and focussing on the needs of this group more in your discussion

Table 2: Given you have performed a binomial logistic regression e.g. with a dichotomous outcome variable not multinomial logistic regression, it feels misleading to show success vs. abandonment vs. failure vs. death in Table 2, consider formatting Table 2 with favourable vs. unfavourable outcomes. Please amend ‘IC95%’ to ’95% CI’. Consider rearranging the reference group to first sub-group for every covariate of interest. Please avoid using commas as decimal points. Please include ‘Ref.’ in your figure legend. Do you think that the ‘self-identified ethnicity’ variable is limited in its use here?

- Discussion

Suggest condensing paragraph one and focussing more on your own results and placing them into context, e.g. in the region and then globally

‘Notably, the diagnosis and treatment of MDR/RR-TB in Colombia is free for all patients regardless of health regime affiliation’ – so actually we are seeing the impact of hidden costs of treatment here? Worth clarifying what is and is not included in your treatment if you are in the subsidised group. Would be helpful to include some background as to catastrophic health spending in Colombia for patients with TB, or more broadly (https://www.researchgate.net/publication/50228851_Determining_factors_of_catastrophic_health_spending_in_Bogota_Colombia)

‘At the national level, incidence and mortality rates, adjusted for age and sex, were higher for the subsidized regime in 37 events of public health interest’ – please clarify what these 37 events are? And how meaningful is this if incidence and mortality has only been adjusted by age and sex.

‘Other countries such as Nigeria also documented a higher rate of successful TB treatment among patients treated in the private sector’ – however being treated in a public healthcare facility was not independently associated with unfavourable treatment outcome in your multivariate analysis– so I think this comparison may be limited. Are there any other comparable studies in South America?

‘The care received at first-level HCIs (health centers) was a protective factor for unfavorable outcomes’- Again this did not persist on adjustment in your multivariate model, and it could be worth exploring this further in your discussion

Please consider using sub-headings in the discussion to clearly structure this section

- Conclusions

Please consider rephrasing ‘This high rate of unfavourable treatment outcomes was associated with affiliation with the subsidized health system regime and age ≥ 60 years.’ - e.g. could be adapted to: Those who qualified for subsided health care or who were aged over 60 years were independently more likely to experience unfavourable treatment outcomes

Reviewer #2: Dear Authors, the study “Factors associated with unfavorable treatment outcomes in patients with rifampicin-resistant tuberculosis in Colombia” has used a nationwide data on drug resistant TB in Colombia and the analysis are adequate for the research question. Nevertheless, the methods and results should be better described to evaluate some important aspects of data analysis. Also, the discussion could include a better literature review between poverty and unfavourable treatment outcomes, as the main study result show that people that are poor and therefore are being treated in the public healthcare sector in the country are less likely to complete treatment. Please see the suggestion below:

Major points

1. In page 5 the authors make a comprehensive description of Colombian health system and describe that all TB treatment is provided for free through the Colombian government (TB programme). Nevertheless, later they include a variable that divide health regime in “subsidized” or “contributive”. I think the authors maybe should make more clear (if what I understood is correct) that independent of the type of health regime, after being diagnosed, the treatment is provided for free. Also, it would be nice to know if this measure is more likely to a proxy of socioeconomic status than the treatment provision itself.

2. In page 6/7, study type and population, please describe the inclusion and exclusion criteria.

3. In page 7, row 126/127: the sentence is confusing. Maybe only say “seventy patients with unknown treatment outcome were excluded”? Also, this also should be moved to the results section.

4. In the methods section, it would be important to include a subsection describing the variables that were extracted from the TB dataset and with which purpose.

5. In page 7, row 125-133: what method was used to include variables in the adjusted model? Backward, forward or the variables were included all at once? The authors considered any confounding variable.

6. In page 7, row 125-133: Some variables included in the results section were not described in the methods. Please include a general description of variables in the adequate section

7. In page 7, row 125-133: how missing data was handled in the analysis?

8. In the methods section, the definition of terms should be included before the statistical analysis.

9. In page 8, row 156-175: the authors are repeating the definitions of treatment outcomes multiple times in the paragraph. Maybe try to be briefer for the readers to follow.

10. In page 8, row 171-175: outcomes not evaluated were included or excluded from the analysis? This should be clear in the methods.

11. In page 9, descriptive analysis, it would be important to describe if there was an increase or decrease in the number of RR-TB over time.

12. In page 9, descriptive analysis, the authors could describe the percentage of cases newly diagnosed and those who are being retreated.

13. In table 1 and 2, the authors included the diagnosis method as a possible explanatory variable for unfavourable treatment outcomes. What is the hypothesis behind including such a variable?

14. Table 1 is secondary for the authors research question. Maybe it should be moved to supplementary material.

15. In Table 2, please describe which ethnicities were included in the “other” category.

16. Figure 2 is not very adequate to show what the authors meant. Maybe a simple one bar graph for each city with the proportions of favourable vs unfavourable outcomes and the N on the top would be clearer. Also, if possible, it would be relevant to know the proportion of RR-TB among all TB cases in each city once the N of RR-TB depends on the number of overall TB cases and the population of each city.

17. In page 13, row 230: did the authors tested for trend in the relationship between age and lower treatment success?

18. In page 17, row 269-271: this sentence is confusing.

19. In the discussion section, the authors make a very important comparison of the study results with other Colombian literature, but have made a limited comparison with the literature from other LMIC countries or elsewhere. For example, if being treated in the private sector is a proxy of wealth, which other studies have shown an association between wealth and poor treatment outcomes?

20. In the limitation section, absence of further socioeconomic data should be included as a limitation for controlling for confounder in the analysis. There is a vast literature showing the relationship between poverty and unfavourable treatment outcomes, as well the effect of poverty reduction measures on improving TB treatment (PMID: 31000126, PMID: 30740248, PMID: 26884501).

21. In the conclusion, the authors should try to demonstrate how the study can be used for policy making or for improving health of those most needed or for improving treatment, or which further research questions are needed.

22. The paper should be revised by a native English speaker.

Minor points

1. In the abstract, it is important to make it more clear for the readers what “affiliated with the health regime” means.

2. In page 4, rows 57 – why these conditions represent public health problems?

3. In page 5, row 78 – the word complex does not mean much. The system is complex or the authors meant something else? it would be good to use a more specific word.

4. In page 7, row 127: maybe is more accurate to say descriptive analysis instead of frequency analysis?

5. In page 7, row 130: A logistic regression model was used to analyze the “association” and not the “interaction” between variables.

6. In page 8, row148-153: Please describe all the full names before using abbreviations for the first time (eg. RMP, INH and please check in the text.)

6. PLOS authors have the option to publish the peer review history of their article (what does this mean?). If published, this will include your full peer review and any attached files.

Reviewer #1: **Yes: **Louisa Chenciner

Reviewer #2: **Yes: **Julia M Pescarini

---

## [Author Response · Author response to Decision Letter 0]

30 Sep 2020

Dr. Tom E. Wingfield

Academic Editor

PLOS ONE Response to Reviewers

Ref: PONE-D-20-18648

Factors associated with unfavorable treatment outcomes in patients with rifampicin-resistant tuberculosis in Colombia

Dear Dr. Tom E. Wingfield

We then respond to questions and suggestions from the editor and reviewers. We look forward to meeting the requirements and expectations of PLOS ONE. We are available for further questions or suggestions about this work

OK.

Please can you systematically address the reviewers' comments paying special attention to:

1) Enhancing the literature review and introduction with relation to social determinants of TB and their further association with adverse TB treatment outcomes

2) Clarify some parts of the methods including: inclusion and exclusion criteria; evidence informing inclusion of independent variables into your regression model (e.g. method of diagnosis); and methods used to arrive at your adjusted model

3) Improvements to the written text and flow

4) Adjustments to the Tables and Figures (see Reviewer 2's comments) to ensure that only the most relevant and applicable information is included and the tables and figures are supportive of the main study objectives.

We will systematically address the reviewers' comments paying special attention to these four items

Reviewer #1

Major Revisions

1. Please mention clearly that you first performed bivariate analysis, and whether thereafter included variables with p value < 0.05

Line: 130 to 134 “Bivariate analysis: A logistic regression model was used to analyze the interaction of the exposure variables (age, sex, ethnicity, site of TB, health insurance scheme, treating HCI, level of care, method used for diagnosis and type of resistance) with the response variable (treatment outcome). Variables with p-values < 0.05 in the bivariate analysis were considered for the multivariate analysis”

2. Please clarify whether, and which, variables were included or excluded a priori from the multivariate analysis to clarify the model building process

Line: 133 to 135 “Variables with p-values < 0.05 in the bivariate analysis were considered for the multivariate analysis. We did not include or exclude a priori variables from the multivariate analysis”.

3. Please specify which confounders were included in the analysis here, and which (if any) registry variables were excluded. 

A possible confounding variable included in the analysis is the health institution (public or private). We do not exclude registry variables.

4. Were any other variables excluded due to missing data? Would be helpful to specify % of missing data here

Other variables were not excluded, the percentage of data lost was 13.7% 

Line: 194 “and 70 (13,7%) patients were excluded because treatment outcomes was unknown”

5. Figure 1: Can you clarify why you decided to exclude 396 individuals with INH resistant TB? Did you perform a separate analysis which included this group to see how outcomes differed?

Only rifampicin-resistant patients were included in this study. Because in Colombia the treatment is different: those resistant to rifampicin receive second-line treatment that includes fluoroquinolones. Patients with isionezide resistance receive R E Z daily for 9 months.

A separate analysis was performed that is not included in this publication, where it was observed that the results of treatment in patients with isoniazid resistance are very similar to those with drug-sensitive tuberculosis.

6. Table 1: How and why did you choose to disaggregate age into <20, 20 to 39, 40 to 59 and over 60 – it appears the majority are in the 20-59 bracket and valuable information may be lost by grouping into such large categories. Did you perform a sensitivity analysis using smaller age intervals?

We arrived at this age categorization, after conducting sensitivity analysis, desegregating the variables at intervals of every 5 years and every 10 years.

In this cohort, we noted that there were few patients younger than 20 years and older than 60 years, and in each patient the results of the treatment behaved differently. While most of the patients were between 20 and 59 years old. This group was disaggregated into 2 groups almost equal in number of patients, trying to create comparable groups without losing much information.

7. There are only 16 patients with XDR TB; are you convinced this sample is large enough to make meaningful conclusions about the rate of unfavourable treatment outcomes in this group?

16 patients is a small n to draw meaningful conclusions, and this could be a devil in this study. however, in Colombia, an average of 5 XDR cases are diagnosed and reported per year. and 16 patients are the total of patients diagnosed and reported with XDR in the 3 years of study. Therefore, studies that include long periods are recommended, because the incidence of XDR is not very high in Colombia.

8. Please rationalise why you have included method of diagnosis in your analysis, how do you anticipate this would be associated with treatment outcome?

The diagnostic method of rifampicin resistance was included for a better description of the dynamics of the diagnosis and treatment of RR TB in Colombia; we did not expect it to influence the treatment results. Which was corroborated in the bivariate analysis and did not enter the multivariate analysis.

9. Consider using primary healthcare as reference group in this analysis; it seems unsurprising here that primary healthcare is a ‘protective factor’ as surely this variable is just a proxy for patients being ‘less sick’. I would be more interested to see whether those receiving secondary/tertiary level care had better or worse outcomes – does being in a tertiary centre result in more specialised care or are patients sicker?

I agree that the best treatment results among patients seen in the first level of care may be related to the severity of the disease.

We consider using first-level care as a reference. We found that being seen at the second or third level could be associated with unfavorable results; however, this association was not significant.

Line 257 to 258 “In turn, being seen in a second or third level HCIs was associated with unfavorable results, although the association was not significant (ORc = 1.6; 95% CI: 1.0 - 2.5)”

10. ‘The strong association between health regime and TB treatment outcomes in Colombia is relevant, as individuals affiliated with the government-subsidized health regime, i.e., the poorest and most vulnerable population, had a higher probability of presenting unfavourable treatments’ 

Line 274 to 277 “The association between the health regimen and the results of TB treatment in Colombia is relevant, since individuals affiliated with the government-subsidized health regimen, which are people with low economic resources, without the ability to pay contributions to the health system, were more likely to present unfavorable treatment results”

11. and ‘The conditions of poverty and vulnerability of the population affiliated with the subsidized regime may be associated with different barriers accessing basic and health services’ - This feels like a leap; given your model does not include household income, education level, living conditions – isn’t qualifying for subsidised healthcare a proxy for socioeconomic deprivation in this analysis? It may be difficult to confirm whether the discrepancy in treatment outcome for this group relates to barriers to access, quality of care, or other social factors for unfavourable outcome. Consider rephrasing and discussing these limitations more fully.

Line 285 to 290 “Establishing why the population affiliated with the subsidized health regime is more likely to have unfavorable TB treatment outcomes is beyond the scope of this study. Since we do not have enough socioeconomic variables such as family income to analyze in this cohort. Where previous studies have suggested that economic barriers, including transportation, medication, and examination costs, or geographic barriers, cause these individuals to have limited access to health care”.

12. ‘Therefore we carry out a careful adjustment of possible confounding factors, seeking to reduce biases’- what kind of bias did you seek to reduce, and what other confounders should have been included? Please explain how your results may have been affected (e.g. over/underestimation of odds ratios) by these confounders being omitted

Line 322 to 331 “so we carried out a multivariate analysis that included all the factors that were associated with unfavorable treatment outcomes in the bivariate analysis, seeking to highlight the factors that actually influence TB treatment outcomes and to rule out confounding factors. However we feel limited the possibility of studying other factors that could influence the outcomes of treatment for MDR/RR-TB, such as coinfection with HIV, comorbidities such as diabetes and habits such as alcohol, tobacco, psychoactive substance use and socioeconomic factors such as family income, as these variables were not considered in the data provided. The absence of these variables could lead to an overestimation of the association between health status and TB treatment outcomes”

13. Try to align the conclusions more closely with your results and discussion

Line 346 to 348 “Patients affiliated to the subsidized health regime were 3 times more likely to present unfavorable results than those affiliated to the contributory health regime, and age over 60 was also associated with unfavorable results”

Minor Revisions

- Title

1. Could be tweaked to correspond directly to its contents, there is some interchangeable reference between MDR-TB, DR-TB, XDR-TB, RR-TB and RMP. Would also be good to specify that it is a retrospective cohort study and registry based in the title

“Factors associated with unfavorable treatment outcomes in patients with rifampicin-resistant tuberculosis in Colombia 2013 - 2015 a retrospective cohort”

- Abstract

2. Background: could you clarify what you mean by ‘exerts great pressure on the complex Colombian health system’, e.g. interesting to know whether you are referencing treatment costs, lengthy treatment, inadequate infrastructure?

Line 80 to 82 “MDR/RR-TB requires prolonged and expensive treatment, which is difficult to sustain in a Colombian health system that requires the joint action of different actors to provide health services”.

3. Methodology: If able please specify the study inclusion criteria, and that the study uses registry level data in the abstract. Are you including those who have received previous treatment?

A retrospective cohort study was conducted including all patients who initiated treatment for MDR/RR-TB between January 2013 and December 2015 in Colombia, who were registered and followed up by the national TB control program.

4. Results: Language used in the results is inconsistent with the methodology in the abstract, would be helpful to review the sentences ‘511 patients who started treatment for MDR/RR-TB’ and ‘A total of 511 patients were diagnosed with RR-TB in Colombia’. Suggest reviewing the language ‘affiliated with the subsidized health regime’, for example could state ‘individuals who qualified for subsidised health care services’

Line 40 to 41 “a total of 511 patients with MDR/RR-TB were registered and followed up by the national TB control program in Colombia”

“affiliated with the subsidized health regime” we consider that it is the term that best describes the characteristic

5. Consider making it clearer in the abstract whether you included all patients with DR-TB in your regression model, or whether you performed sub-group analysis comparing MDR-TB, RMP and XDR-TB

 Line 48 to 52 “The 511 MDR/RR-TB patients were included in the bivariate and multivariate analysis, identifying that the age ≥ 60 years (crude odds ratio (ORc) = 2.4, 95% CI 1.1 – 5.8; adjusted odds ratio (ORa) = 2.7, 95% CI 1.1 – 6.8) and affiliation with the subsidized health regime (ORc = 3.6, 95% CI 2.3 – 5.6; ORa = 3.4, 95% CI 2.0 – 6.0) were associated with unfavorable treatment outcomes”.

6. Suggest reviewing key words to ensure they relate more closely to your paper, e.g. consider dropping ‘associated factors’

Line 59 to 60 “Tuberculosis, MDR/RR-TB, MDR-TB, treatment outcomes, unfavorable treatment outcomes”

Introduction

7. Suggest rephrasing ‘these conditions represent public health problems around the world’, and relating more directly to the challenge that DR-TB poses

Line 66 to 69 “These conditions are generally the consequence of social and political decisions that lead to inadequate compliance with the DOTs strategy (strictly supervised shortened treatment) by government entities, sociodemographic barriers that prevent access to medicines (e.g. living in rural areas, not having health insurance)”

8. Line 61 – are these definitely marked ‘improvements’?

Line 74 to 76 “Although more people underwent treatment in 2018, they accounted for only 32% of the estimated incidence, and the treatment success rate was 56% for MDR/RR-TB and 39% for XDR-TB”

9. Please ensure that in-text citations are inside of the sentence, e.g. before full stop.

We make sure we put the points in the right place

10. ‘In 2018, 205 67 cases were notified out of an estimated 580, for a detection rate of 35.3%, similar to 68 the detection rate of 32% reported worldwide’ – please clarify which cases you are referring to; it sounds like you suspect there is underreporting of DR-TB in Colombia?

Line 82 to 83 “In 2018, 205 cases of MDR/RR-TB were notified out of an estimated 580” 

We refer to cases of MDR/RR - TB.

We really believe that there is underreporting of RR-TB cases, but it is more worrying to think that RR-TB cases are not detected in a timely manner. However, we have no evidence to support this statement.

11. Several references to ‘pressure’ and the ‘complex Colombian health System’ – please clarify in which respect, e.g. are you referencing lack of resources, infrastructure, workforce? Suggest maintaining consistent language, e.g. stick with treatment, not mentioning pharmacotherapy in the study objective

The word Pressure was reformulated in the text and "complex health system" refers to the fact that the Colombian health system has three important actors, the government as a control entity, the companies that administer health services and the companies that provide health services. for a person to receive health care it is necessary that these three actors converge in favor of the individual and thus guarantee the provision of health services. For better understanding, this phrase was also substituted in the text.

Methods

Study Site

12. You may be able to better summarise or condense the provision of healthcare in Colombia by using a figure or schematic

We believe it is possible; however, the authors prefer to explain it in the text.

13. Please confirm whether subsidised healthcare includes additional support, e.g. social protection measures

Line 120 to 122 “However, people who qualified for subsidized health care services are not necessarily beneficiaries of social protection programs”

14. Suggest paragraph 1 e.g. line 89-93 could be better summarised and more clearly convey the importance of understanding DR-TB in this study setting

Line 111 to 113 “In this country of marked inequality, the social security system in health presents difficulties in responding to the economic and social demands involved in the treatment of MDR/RR-TB”.

15. Could be helpful to mention earlier that the direct costs of treatment are ‘free’ at point of care, but there are hidden costs. Also please confirm that TB is a notifiable disease in Colombia.

Line 136 to 138 “In summary, in Colombia TB is a disease of public health interest. Is of obligatory notification, its diagnosis, treatment and follow-up are covered by public resources”.

Study Type and Population

16. Please consider specifying whether the dataset was anonymised or de-identified

The dataset was anonymised

17. Helpful to provide exact dates of study time frame, given you later state it might have been too early for treatment outcomes to be recorded for those where its treatment outcome was missing. Please confirm when treatment outcome was recorded (e.g. time to follow-up), and how

Line 141 to 145 “A retrospective cohort study was conducted that included all patients who were notified and diagnosed with MDR/RR-TB, with the start of treatment between January 2013 and December 2015 in Colombia. Data on treatment results were collected until December 2017, if up to this date no record of treatment results existed they were catalogued as missing”.

18. Notification and diagnosis are used interchangeably, helpful to stick to one term only as these can mean slightly different things

We will not indicate at the end of the notification

Statistical Analysis in Methodology

18. Reconsider the language ‘analyze the interaction’, this does not really reflect the statistical analysis reported in the results – unless you did examine interaction terms? Otherwise suggest ‘to estimate the association between…’

Line 160 “association between the exposure variables”

19. Please specify which confounders were included in the analysis here, and which (if any) registry variables were excluded

Was answered in question 3 of the major reviews

Definition of Terms

20. Specify what you mean by the ‘Colombian National Policy’

It refers to the guidelines for programmatic management of tuberculosis in Colombia. The phrase was replaced in the text.

Line 191 to 192 “guidelines for management tuberculosis and leprosy program in Colombia”

Results

Descriptive Analysis

21. Figure 2: Is it necessary to show rates of both favourable and unfavourable treatment outcomes on the same graph? Please include y axis for scale. Consider excluding this figure, I am not sure that it helps to answer the study objective.

The figure will be excluded

22. What is the derivation of ‘afro-descendant’? Perhaps consider changing to Afro-Colombian. Who comprises the ‘other’ group – it seems this is the ‘majority’ – How has ethnicity been defined here?

In Colombia, Afro-descendant is synonymous with Afro-Colombian. 

The group called others refers to the mestizos who in Colombia are the majority group

We will change the name to Afro-Colombians and mestizos

23. Reconsider phrasing ‘first’, ‘second’ and ‘third’ care as ‘primary’, ‘secondary’ and ‘tertiary’

The change was made

24. On line 211, please reconsider ‘were affiliated with the subsidized social security and health regime’. Does the subsidised health care also include social protection and other welfare benefits? This contradicts the earlier definition o Treatment outcomes and factors associated with Unfavourable outcomes

Line 244 “were affiliated with the subsidized health regime”

25. Describing ‘the treatment success rate with respect to age showed an inversely proportional trend’ seems like a bit of a leap; could be worth rephrasing, and focussing on the needs of this group more in your discussion

Line 262 to 264 “The older age was associated with a lower success rate, but only patients aged ≥ 60 years showed a significant association with unfavorable outcomes”

26. Table 2: Given you have performed a binomial logistic regression e.g. with a dichotomous outcome variable not multinomial logistic regression, it feels misleading to show success vs. abandonment vs. failure vs. death in Table 2, consider formatting Table 2 with favourable vs. unfavourable outcomes. Please amend ‘IC95%’ to ’95% CI’. Consider rearranging the reference group to first sub-group for every covariate of interest. Please avoid using commas as decimal points. Please include ‘Ref.’ in your figure legend. Do you think that the ‘self-identified ethnicity’ variable is limited in its use here?

The suggestion is accepted and table 2 is modified

Discussion

27. ‘Notably, the diagnosis and treatment of MDR/RR-TB in Colombia is free for all patients regardless of health regime affiliation’ – so actually we are seeing the impact of hidden costs of treatment here? Worth clarifying what is and is not included in your treatment if you are in the subsidised group. Would be helpful to include some background as to catastrophic health spending in Colombia for patients with TB, or more broadly (https://www.researchgate.net/publication/50228851_Determining_factors_of_catastrophic_health_spending_in_Bogota_Colombia)

Line 316 to 320 “Notably, the diagnosis and treatment of MDR/RR-TB in Colombia is free for all patients regardless of health regime affiliation.[14] However there is a differential attention between the two health affiliation regimes, which can be called organizational barrier, which is a factor that hinders the initial contact with the health services (entry barriers) and, also, the timely attention after the patient enters the health center (interior barriers).”

Line 337 to 343 “Also the patients always incur out-of-pocket expenses, which may be direct, such as those related to transportation and examinations or consultations in addition to the provisions of the TB control program, or indirect, such as inability to work due to the disease [24]. These expenses in subsidized families without the ability to pay can create significant barriers to both access and permanence in the TB control program. In this sense, a study carried out in Bogotá, the capital of Colombia, showed that the lower the household income level, the more likely it was to incur catastrophic health spending [25]”. 

28. ‘At the national level, incidence and mortality rates, adjusted for age and sex, were higher for the subsidized regime in 37 events of public health interest’ – please clarify what these 37 events are? And how meaningful is this if incidence and mortality has only been adjusted by age and sex.

Line 321 to 328 “Inequalities related to the health system affiliation regime have been identified previously at the national level, for example for the subsidized regime high incidence and mortality rates were identified, adjusted by age and sex, for events tracing the quality of health care, such as mortality in children under five years of age due to acute respiratory infection, acute diarrheal disease, and malnutrition; events related to sexual and reproductive health, such as maternal mortality, gestational syphilis, and congenital syphilis; infectious diseases, such as leishmaniasis, Chagas' disease, and malaria; and poverty-related communicable diseases, such as leprosy and tuberculosis.

29. ‘Other countries such as Nigeria also documented a higher rate of successful TB treatment among patients treated in the private sector’ – however being treated in a public healthcare facility was not independently associated with unfavourable treatment outcome in your multivariate analysis– so I think this comparison may be limited. Are there any other comparable studies in South America?

Suggestion accepted, the comment is removed from the text

30. ‘The care received at first-level HCIs (health centers) was a protective factor for unfavorable outcomes’- Again this did not persist on adjustment in your multivariate model, and it could be worth exploring this further in your discussion

Line 358 to 359 “Care in second and third level health facilities was associated with unfavorable outcomes”

- Conclusions

31. Please consider rephrasing ‘This high rate of unfavourable treatment outcomes was associated with affiliation with the subsidized health system regime and age ≥ 60 years.’ - e.g. could be adapted to: Those who qualified for subsided health care or who 

were aged over 60 years were independently more likely to experience unfavourable treatment outcomes

Line 391 to 395 “In conclusion, in Colombia, 50.1% of the patients with RR-TB who initiated treatment between January 2013 and December 2015 showed unfavorable treatment outcomes, and 19.7% died during treatment. Patients with XDR-TB had unfavorable outcomes in 85.7% of cases and with MDR-TB in 47.6% of cases. Patients who qualified for subsidized care or were over 60 years old were more likely to experience unfavorable treatment outcomes independently more likely to experience unfavourable treatment outcomes”

Reviewer #2

1. In page 5 the authors make a comprehensive description of Colombian health system and describe that all TB treatment is provided for free through the Colombian government (TB programme). Nevertheless, later they include a variable that divide health regime in “subsidized” or “contributive”. I think the authors maybe should make more clear (if what I understood is correct) that independent of the type of health regime, after being diagnosed, the treatment is provided for free. Also, it would be nice to know if this measure is more likely to a proxy of socioeconomic status than the treatment provision itself.

Modifications were made to clarify this information in both the methods and the discussion: line 121 to 124 “and the subsidized regime, which covers the remaining 53%,[10] sheltering all poor and vulnerable people, for whom health services are covered by government resources.[11] However, people who qualified for subsidized health care services are not necessarily beneficiaries of social protection programs”.

Line 136 to 138 “In summary, in Colombia TB is a disease of public health interest. Is of obligatory notification, its diagnosis, treatment and follow-up are covered by public resources”

Line 316 to 320 “Notably, the diagnosis and treatment of MDR/RR-TB in Colombia is free for all patients regardless of health regime affiliation.[14] However there is a differential attention between the two health affiliation regimes, which can be called organizational barrier, which is a factor that hinders the initial contact with the health services (entry barriers) and, also, the timely attention after the patient enters the health center (interior barriers)”.

2. In page 6/7, study type and population, please describe the inclusion and exclusion criteria.

Line 141 to 145 “that included all patients who were notified and diagnosed with MDR/RR-TB, with the start of treatment between January 2013 and December 2015 in Colombia. Data on treatment results were collected until December 2017, if up to this date no record of treatment results existed they were catalogued as missing data”.

3. In page 7, row 126/127: the sentence is confusing. Maybe only say “seventy patients with unknown treatment outcome were excluded”? Also, this also should be moved to the results section.

Line 144 to 145 “if up to this date no record of treatment results existed they were excluded, catalogued as missing data”.

Line 224 to 225 “and 70 (13,7%) patients were excluded because treatment outcomes was unknown”

4. In the methods section, it would be important to include a subsection describing the variables that were extracted from the TB dataset and with which purpose.

No variables were extracted, we worked with all the variables available in the database provided by colombia's TB control program

5. In page 7, row 125-133: what method was used to include variables in the adjusted model? Backward, forward or the variables were included all at once? The authors considered any confounding variable.

Line 163 to 165 “Variables with p-values < 0.05 in the bivariate analysis were considered for the multivariate analysis. We did not include or exclude a priori variables from the multivariate analysis”.

6. In page 7, row 125-133: Some variables included in the results section were not described in the methods. Please include a general description of variables in the adequate section

Line 159 to 162 “Bivariate analysis: A logistic regression model was used to analyze the association between interaction of the exposure variables (age, sex, ethnicity, site of TB, Health regimehealth insurance scheme, treating HCI, level of care, method used for diagnosis and type of resistance) with the response variable (treatment outcome).

7. In page 7, row 125-133: how missing data was handled in the analysis?

Missing data were excluded from the analysis

8. In the methods section, the definition of terms should be included before the statistical analysis.

Suggestion accepted the change was made

9. In page 8, row 171-175: outcomes not evaluated were included or excluded from the analysis? This should be clear in the methods.

Line 141 to 145 “A retrospective cohort study was conducted that included all patients who were notified and diagnosed with MDR/RR-TB, with the start of treatment between January 2013 and December 2015 in Colombia. Data on treatment results were collected until December 2017, if up to this date no record of treatment results existed they were excluded, catalogued as missing data”.

10. In page 9, descriptive analysis, it would be important to describe if there was an increase or decrease in the number of RR-TB over time.

Line 233 to 234 “167 cases were diagnosed in 2013, 181 in 2014 and 163 in 2015”.

11. In page 9, descriptive analysis, the authors could describe the percentage of cases newly diagnosed and those who are being retreated.

Sorry, but we do not have that variable in our database

12. In table 1 and 2, the authors included the diagnosis method as a possible explanatory variable for unfavourable treatment outcomes. What is the hypothesis behind including such a variable?

The diagnostic method of rifampicin resistance was included for a better description of the dynamics of the diagnosis and treatment of RR TB in Colombia; we did not expect it to influence the treatment results. Which was corroborated in the bivariate analysis and did not enter the multivariate analysis.

13. Table 1 is secondary for the authors research question. Maybe it should be moved to supplementary material.

Accepted suggestion

14. In Table 2, please describe which ethnicities were included in the “other” category.

The term others was replaced by Mestizo which is the majority race in Colombia

15. Figure 2 is not very adequate to show what the authors meant. Maybe a simple one bar graph for each city with the proportions of favourable vs unfavourable outcomes and the N on the top would be clearer. Also, if possible, it would be relevant to know the proportion of RR-TB among all TB cases in each city once the N of RR-TB depends on the number of overall TB cases and the population of each city.

This figure was removed from the document at the suggestion of reviewer #1

16. In page 13, row 230: did the authors tested for trend in the relationship between age and lower treatment success?

Line 275 to 277 “older age was associated with a lower success rate, but only patients aged ≥ 60 years showed a significant association with unfavorable outcomes (ORc = 2.4, 95% CI 1.1 - 5.8; ORa = 2.7, 95% CI 1.1 - 6.8)”.

17. In the discussion section, the authors make a very important comparison of the study results with other Colombian literature, but have made a limited comparison with the literature from other LMIC countries or elsewhere. For example, if being treated in the private sector is a proxy of wealth, which other studies have shown an association between wealth and poor treatment outcomes? In the limitation section, absence of further socioeconomic data should be included as a limitation for controlling for confounder in the analysis. There is a vast literature showing the relationship between poverty and unfavourable treatment outcomes, as well the effect of poverty reduction measures on improving TB treatment (PMID: 31000126, PMID: 30740248, PMID: 26884501).

Line 356 to 360 “In Colombia, qualifying to be a beneficiary of the subsidized health regime is synonymous with belonging to the poorest and most vulnerable population, and TB has a direct relationship with poverty and social exclusion. For example, in Brazil, they concluded that being a beneficiary of the Bolsa Familia Program (BFP) was an independent factor that influenced the favorable outcome of people receiving drug treatment for TB”.

18. The paper should be revised by a native English speaker.

Suggestion accepted

American Journal Experts was hired to perform the English translation and style correction of the manuscript

Minor points

1. In the abstract, it is important to make it more clear for the readers what “affiliated with the health regime” means.

Line 121 to 124 “and the subsidized regime, which covers the remaining 53%,[10] sheltering all poor and vulnerable people, for whom health services are covered by government resources.[11] However, people who qualified for subsidized health care services are not necessarily beneficiaries of social protection programs”.

2. In page 4, rows 57 – why these conditions represent public health problems?

Line 65 to 69 “These conditions are generally the consequence of social and political decisions that lead to inadequate compliance with the DOTs strategy (strictly supervised shortened treatment) by government entities, sociodemographic barriers that prevent access to medicines (e.g. living in rural areas, not having health insurance”

3. In page 5, row 78 – the word complex does not mean much. The system is complex or the authors meant something else? it would be good to use a more specific word.

Line 94 to 96 “MDR/RR-TB requires prolonged and expensive treatment, which is difficult to sustain in a Colombian health system that requires the joint action of different actors to provide health services”

4. In page 7, row 127: maybe is more accurate to say descriptive analysis instead of frequency analysis?

Line 210 to 211 “A descriptive analysis was performed”

5. In page 7, row 130: A logistic regression model was used to analyze the “association” and not the “interaction” between variables.

Line 212 to 213 “Bivariate analysis: A logistic regression model was used to analyze the association between the exposure variables”

6. In page 8, row148-153: Please describe all the full names before using abbreviations for the first time (eg. RMP, INH and please check in the text.)

Suggestion accepted

Line 62 to 63 “Multidrug resistant tuberculosis (MDR-TB) is defined as TB resistant to at least rifampicin (RMP) and isoniazid (INH), and extensively resistant TB (XDR-TB)”

Thank you for your consideration!

Sincerely,

Ninfa Marlen Chaves Torres

Professor, School of Medicine 

Nueva Granada Military University

---

## [Decision Letter · Decision Letter 1]

17 Nov 2020

PONE-D-20-18648R1

Factors associated with unfavourable treatment outcomes in patients with rifampicin-resistant tuberculosis in Colombia 2013 - 2015 a retrospective cohort

PLOS ONE

Dear Dr. Chaves Torres,

Thank you for submitting your manuscript to PLOS ONE. After careful consideration, we feel that it has merit but does not fully meet PLOS ONE’s publication criteria as it currently stands. 

The reviewers agreed that the manuscript was much improved but there remain areas that still require attention. I would be grateful if you could review both of the reviewers' comments and refine your manuscript accordingly and we then invite you to re-submit for further review.

We look forward to receiving your revised manuscript.

Kind regards,

Tom E. Wingfield

Academic Editor

PLOS ONE

Reviewers' comments:

Reviewer's Responses to Questions

**Comments to the Author**

1. If the authors have adequately addressed your comments raised in a previous round of review and you feel that this manuscript is now acceptable for publication, you may indicate that here to bypass the “Comments to the Author” section, enter your conflict of interest statement in the “Confidential to Editor” section, and submit your "Accept" recommendation.

Reviewer #1: (No Response)

Reviewer #2: All comments have been addressed

2. Is the manuscript technically sound, and do the data support the conclusions?

Reviewer #1: Partly

Reviewer #2: Yes

3. Has the statistical analysis been performed appropriately and rigorously? 

Reviewer #1: Yes

Reviewer #2: No

4. Have the authors made all data underlying the findings in their manuscript fully available?

Reviewer #1: Yes

Reviewer #2: Yes

5. Is the manuscript presented in an intelligible fashion and written in standard English?

Reviewer #1: No

Reviewer #2: No

6. Review Comments to the Author

Reviewer #1: I would like to make the following suggestions:

Major Revisions:

1. Language

- Paper, including abstract, needs revision by native English speaker

- Avoid value laden language: examples including lines 60-64 ‘these conditions are generally the consequence’ – implies causality, line 99 ‘marked inequality’, line 109 ‘sheltering all poor and vulnerable people’ – does this policy definitely shelter all?, line 285 ‘Also the patients always incur out-of-pocket expenses’, line 289 ‘in this sense’, line 323 ‘actually influence’, line 324 ‘to rule out confounding factors’, line 334 ‘we believe’

- Aim for consistency in language/terminology used: examples include referencing ‘poor treatment outcome’ vs. ‘unfavourable treatment outcome’; ‘cases’ vs. ‘patients’ in conclusion paragraph

- Aim for clear language: examples including lines 221-222 ‘departmental capitals’, concerned that line 134 ‘individualised and anonymous information’ is a contradiction in terms, line 274 ‘differential attention’, line 289 ‘permanence in TB control’, line 233-234 consider rephrasing ‘the older age was associated with a lower success rate, but only patients aged >60 years), line 86-87 ‘different actors’ – are you referencing actors in a health policy setting (e.g. at governmental level) or within the healthcare system?, line 89-90 ‘could help guide the design of national public health strategies’ – could you be more specific here?; line 83 – please clarify which other comorbidities are independently associated with mortality; line 125 ‘obligatory notification’; line 182 ‘logistic regression model was used to analyze’; lines 249-255 ensure distinction is made between odds ratios estimated in bivariate analysis and multivariate analyses; lines 266-267 clarify what you mean by ‘these values’ in the text; consider whether ‘health regime’ or ‘health regimen’ is appropriate use of language

- Avoid potentially stigmatising language relating to TB: example including line 138 ‘treatment abandonment’

- Shorten and condense: examples including lines 279-284 could be condensed further

- Ensure grammar is correct; lines 124-126 – unclear if these are meant to be two separate sentences? ‘In summary, in Colombia TB is a disease of public health interest. Is of obligatory notification, its diagnosis, treatment and follow-up are covered by public resources’; line 324 ‘However we feel limited the possibility of studying’

2. Structure and Flow

- Suggest review of abstract: including: background to ensure this provides clear rationale for the study, study design, and review of lines 42-45 and methods to ensure brief summary of statistical methods used here

- Introduction would benefit from more structured approach, and further review of paragraph order to ensure systematic review is clearly conveyed; Line 78-84 would benefit from revision to distinguish between proximal and distal factors, and to provide context to the variables included latterly in your statistical analysis

- Study site section would benefit from review: both paragraph 1 (lines 95-101) and paragraph 2 (lines 102-120) could be condensed further to clearly convey this information to the reader. Thank you for emphasising that those included in subsidised health services are not necessarily included in other social protection programmes; as reviewer #2 mentioned previously it might be helpful to clarify here or later perhaps whether subsidised health services are therefore a proxy for socioeconomic disadvantage.

- Definition of Terms; could benefit from restructuring of paragraph 1 (lines 136-143) – to ensure clarity for the reader; e.g. first introduce how the Colombian National Program defines drug resistant TB, and secondly why this may be included in the differential diagnosis and how drug resistant TB may be confirmed

-Discussion: I think lines 272-273 could be developed further; e.g. despite free access to MDR/RR-TB, there is discrepancy in treatment outcome and we infer…

-Discussion: paragraph consisting of lines 274-284 - this is important information, but it needs to be more closely related to your study findings and results – and condensed as able; lines 292-296 – there seem to be conflicting arguments presented here – e.g. comparison of public healthcare (without social protection) in Colombia compared to protective effect of Bolsa Familia (social protection) in Brazil; in lines 297-306 please link more closely to your study findings; Consider rephrasing rationale for the analysis used in lines 320-324; Please condense lines 330-333, or divide into two shorter sentences; revise sentence on line 337-338 (several clauses used)

Minor Revisions:

1. Avoid abbreviations if able in abstract, and ensure all abbreviations used are defined in brackets, but also please avoid duplicate definitions – see line 57 and line 66 of MDR TB

2. Would be helpful to know the precise dates for study inclusion criteria – see line 130

3. Could you clarify why data on treatment outcome was only ‘collected until December 2017’ – see line 131?

4. Suggest providing a citation for how treatment outcome was defined – see line 157-158

5. Could you clarify ‘Seventy patients categorized as not evaluated were not included

in the analysis because the treatment outcome was unknown’ – were these missing treatment outcomes, or not yet recorded as a result of the December 2017 cut-off?

6. Thank you for clarifying that you did not include or exclude a priori

7. Suggest formatting throughout (n =; %) – especially noted in lines 216-223; line 229 – ‘87 (19.7)’ need to include ‘%’ for consistency

8. Ensure formatting consistent in Table 1; use decimal points not commas, consider use of bold for reference category/sub-heading; ensure all in English language ‘Menor de 20’

9. Avoid inclusion of odds ratios in discussion (should mostly feature in results)

10. In references, for online resources please provide access/citation date

Reviewer #2: Dear authors, the manuscript improved substantially since the last submission but I believe some further improvements to the abstract and introduction would be desirable so the paper could reach a larger audience. The manuscript would also benefit from a nice public health message. Please see some comments below:

Major

1. The abstract could be revised. The methods section is lacking substantial information on: the variables, outcome definition, statistical methods used for the analysis. Similarly, there is substantial repeated information in the introduction, methods and results. In the conclusions section, the authors could also include some practical application of the results for targeting policies that reduce these health inequalities in Colombia.

2. Introduction - Rows 73-77. This is an interesting paragraph, but hard to follow and not related to the specific objective of the study. I suggest that this bit is removed to give flow to the introduction.

3. Introduction - The authors do not show the readers the current research gaps that this study aim to fulfil.

4. Methods - Page 186, it is still not clear if the authors included all the variables with p<0.05 in the bivariate analysis in the multivariate analysis, without testing for interactions, collinearity, or significance. The authors explained their criteria in the discussion, but it would be clearer in the methods section.

5. Methods- In row 220, why there is 67.8% who were affiliated with the subsidized health regime but only 63.7% were treated in “public HCIs”? Public institutions attend both individuals under the subsidized Individuals and the contributory system? If these variables are very correlated (as explained in the discussion), would not be important to choose only one for the multivariate analysis.

6. In the conclusions section, the last paragraph lack of a public health message that interprets or suggests

7. Please include a "Data Availability Statement"

Minor

7. In the introduction, row 63, when listing the reasons the “and” is missing.

8. In the introduction, Row 69, please replace incidence for either “incident cases”.

9. In row 71, substitute “16,000 estimated cases” by “approximately 16,000 cases”

10. In Table 1, please include the N´s for the columns: “Favourable (N=XX)” and “Unfavourable (N=XX)”

11. Table 1, row “Male”, please change the comma for a point. Check commas used instead of points

12. Figure 1 title, please change “Flow chart” for “flowchart”; Please consider using a title that describes better the figure, e.eg., “Flowchart with selected studied population with MDR/RR-TB in Colombia…”

7. PLOS authors have the option to publish the peer review history of their article (what does this mean?). If published, this will include your full peer review and any attached files.

Reviewer #1: **Yes: **L Chenciner

Reviewer #2: **Yes: **Julia Pescarini

---

## [Author Response · Author response to Decision Letter 1]

31 Dec 2020

Dr. Tom E. Wingfield

Academic Editor

PLOS ONE Response to Reviewers

Ref: PONE-D-20-18648

Factors associated with unfavorable treatment outcomes in patients with rifampicin-resistant tuberculosis in Colombia

Dear Dr. Tom E. Wingfield

A continuación, respondemos a las preguntas y sugerencias del editor y los revisores. Esperamos cumplir con los requisitos y expectativas de PLOS ONE. Estamos disponibles para más preguntas o sugerencias sobre este trabajo

Yours sincerely,

Marlen Chaves 

Reviewer #1

Major Revisions:

1. Language

- Paper, including abstract, needs revision by native English speaker

The document was reviewed by a native English speaker

- Avoid value laden language: examples including lines 60-64 ‘these conditions are generally the consequence’ – implies causality. “These conditions have been associated, among other factors, with social and political decisions that lead to inadequate compliance with the DOT strategy (strictly supervised shortened treatment) by government entities, sociodemographic barriers that impede access to medicines (e.g., living in rural areas, not having health insurance).” 

Line 99 ‘marked inequality’ “In this unequal country, the social security system in health presents difficulties in responding to the economic and social demands involved in the treatment of MDR/RR-TB.”

 Line 109 ‘sheltering all poor and vulnerable people’ – does this policy definitely shelter all? “Housing the majority of poor and vulnerable people” 

Line 285 ‘Also the patients always incur out-of-pocket expenses’ “Also the patients incur out-of-pocket expenses, which may be direct”

Line 289 ‘in this sense’ “A study carried out in Bogotá, the capital of Colombia”

line 323 ‘actually influence’, line 324 ‘to rule out confounding factors’, “so we carried out a multivariate analysis that included all the factors that were associated with unfavorable treatment outcomes in the bivariate analysis, seeking to highlight the factors that truly influence TB treatment outcomes and to discard the confounding factors”

Line 334 ‘we believe’ “However, the results obtained are representative of the Colombian population with MDR/RR-TB because all the cases that received treatment in the country over a period of 3 years were analyzed.”

- Aim for consistency in language/terminology used: examples include referencing ‘poor treatment outcome’ vs. ‘unfavorable treatment outcome’; ‘cases’ vs. ‘patients’ in conclusion paragraph

“85.7% of patients with XDR-TB and 47.6% of patients with MDR-TB had unfavorable results. Patients who were eligible for subsidized care or were over 60 years old were more likely to experience unfavorable treatment outcomes independently”

- Aim for clear language: examples including lines 221-222 ‘departmental capitals “nine of which were capitals”

 Concerned that line 134 ‘individualized and anonymous information’ is a contradiction in terms, “The Ministry of Health and Social Protection of Colombia (MINSALUD) provided the data on the participants in a database with individualized information”

Line 274 ‘differential attention’ “However there are differences in the care provided to patients between the two health affiliations regimes”

Line 289 ‘permanence in TB control’ “These expenses in subsidized families without the ability to pay can create significant barriers to both access to the TB control program and compliance with TB treatment”

Line 233-234 consider rephrasing ‘the older age was associated with a lower success rate, but only patients aged >60 years) “The patients aged ≥ 60 years showed a significant association with unfavorable outcomes”

 Line 86-87 ‘different actors’ – are you referencing actors in a health policy setting (e.g. at governmental level) or within the healthcare system? “which is difficult to sustain in a Colombian health system that requires the joint action of different actors within the same system, to provide health services”

 line 89-90 ‘could help guide the design of national public health strategies’ – could you be more specific here?; “could help guide the design of national public health strategies aimed at improving the care provided to patients with MDR/RR-TB and thus decrease the rate of treatment failure”

Line 83 – please clarify which other comorbidities are independently associated with mortality; “and coinfection with HIV have been specifically associated with death”

Line 125 ‘obligatory notification’ “. Its diagnosis, treatment and follow-up are covered by public resources”

Lines 249-255 ensure distinction is made between odds ratios estimated in bivariate analysis and multivariate analyses; 

It is clear the distinction

Lines 266-267 clarify what you mean by ‘these values’ in the text “These success rates being lower than those reported worldwide in 2015, where the success rate was 55% for cases of MDR-TB and 35% for XDR-TB” 

Consider whether ‘health regime’ or ‘health regimen’ is appropriate use of language: Health regime is the official term for Colombia

- Avoid potentially stigmatizing language relating to TB: example including line 138 ‘treatment abandonment’

The term "treatment abandonment" is part of the definition of suspected MDR TB in Colombia 

- Shorten and condense: examples including lines 279-284 could be condensed further

“for example for the subsidized regime high incidence and mortality rates were identified, adjusted by age and sex, for events tracing the quality of health care, such as mortality in children under five years of age due to acute respiratory infection, maternal mortality, ; and poverty-related communicable diseases, such as leprosy and tuberculosis[23].”

- Ensure grammar is correct; lines 124-126 – unclear if these are meant to be two separate sentences? ‘In summary, in Colombia TB is a disease of public health interest. Is of obligatory notification, its diagnosis, treatment and follow-up are covered by public resources’ “in Colombia TB is a disease of public health interest, its diagnosis, treatment and follow-up are covered by public resources[14].”

 Line 324 ‘However we feel limited the possibility of studying’ “However, one limitation was the impossibility of studying other factors that could influence the outcomes of treatment for MDR/RR-TB”

2. Structure and Flow

- Suggest review of abstract: including: background to ensure this provides clear rationale for the study, study design, and review of lines 42-45 and methods to ensure brief summary of statistical methods used here.

The abstract was reviewed and adjusted as directed by the reviewers

- Introduction would benefit from more structured approach, and further review of paragraph order to ensure systematic review is clearly conveyed; Line 78-84 would benefit from revision to distinguish between proximal and distal factors, and to provide context to the variables included latterly in your statistical analysis. 

The introduction was modified following the reviewers' suggestions, however it was not possible to separate the proximal and distal factors

- Study site section would benefit from review: both paragraph 1 (lines 95-101) and paragraph 2 (lines 102-120) could be condensed further to clearly convey this information to the reader. 

This session was compiled as much as possible in order to give more clarity

- Definition of Terms; could benefit from restructuring of paragraph 1 (lines 136-143) – to ensure clarity for the reader; e.g. first introduce how the Colombian National Program defines drug resistant TB, and secondly why this may be included in the differential diagnosis and how drug resistant TB may be confirmed

“Multidrug-resistant tuberculosis is defined as tuberculosis resistant to isoniazid and rifampicin, with or without resistance to other first-line drugs. The diagnosis of drug-resistant tuberculosis is established only by confirmation of resistance in vitro or by molecular tests to one or more antituberculosis drugs”

-Discussion: I think lines 272-273 could be developed further; e.g. despite free access to MDR/RR-TB, there is discrepancy in treatment outcome and we infer…

“Although the diagnosis and treatment of MDR/RR-TB in Colombia is free for all patients, regardless of their affiliation to the health regime, there are discrepancies in the outcome of treatment and we infer”

-Discussion: paragraph consisting of lines 274-284 - this is important information, but it needs to be more closely related to your study findings and results – and condensed as able; 

“We infer there are differences in the care provided to patients between the two health affiliation regimes, which can be called organizational barrier”

lines 292-296 – there seem to be conflicting arguments presented here – e.g. comparison of public healthcare (without social protection) in Colombia compared to protective effect of Bolsa Familia (social protection) in Brazil; 

“This relationship between poverty and TB was studied in Brazil, where they concluded that the outcome of drug treatment for TB was favorable in the beneficiaries of the Bolsa Familia Program (BFP) was a protective factor against unfavorable treatment outcomes.”

In lines 297-306 please link more closely to your study findings; 

We try to link better with our results

Consider rephrasing rationale for the analysis used in lines 320-324; 

“This study was carried out with secondary data (PNCTB database) obtained from reliable PNCTB records, but with a limited number of variables, so all available variables were analyzed. A multivariate analysis was performed to highlight the factors that really influence the results of the treatment of tuberculosis and to discard the factors of confusion”

Please condense lines 330-333, or divide into two shorter sentences; 

“We also dealt with missing data in 70/511 cases representing a loss of 13.7%, acceptable, the lost patients did not present different characteristics compared to the patients included in the analysis. Could be that at the time of data collection they were still being processed”

Revise sentence on line 337-338 (several clauses used)

“Managing public health databases remains a challenge for countries, Colombia, it already has a detailed database on patients with drug-resistant TB”

Minor Revisions: 

1. Avoid abbreviations if able in abstract, and ensure all abbreviations used are defined in brackets, but also please avoid duplicate definitions – see line 57 and line 66 of MDR TB

We have reviewed the abbreviations

2. Would be helpful to know the precise dates for study inclusion criteria – see line 130

“With the start of treatment between January 1st, 2013 and December 31st, 2015 in Colombia”

3. Could you clarify why data on treatment outcome was only ‘collected until December 2017’ – see line 131?

“Data on treatment results were collected until December 2017, assuming a period of time of at least 2 years, hoping that most patients would manage to finish the treatment”

4. Suggest providing a citation for how treatment outcome was defined – see line 157-158

“Ministerio de Salud y Protección Social, Republica de Colombia. Circular externa 00007 de 2015 [Internet]. Available from: https://www.minsalud.gov.co/sites/rid/Lists/BibliotecaDigital/RIDE/DE/DIJ/circular-externa-0007-de-2015.pdf”

5. Could you clarify ‘Seventy patients categorized as not evaluated were not included

in the analysis because the treatment outcome was unknown’ – were these missing treatment outcomes, or not yet recorded as a result of the December 2017 cut-off?

The 70 patients who were not included in the analysis did not have their treatment outcome recorded in the database provided by Colombia's NTP. It may be that 2 years after treatment began, they had not yet finished the treatment, or the result of the treatment was simply not known due to loss to follow-up.

8. Ensure formatting consistent in Table 1; use decimal points not commas, consider use of bold for reference category/sub-heading; ensure all in English language ‘Menor de 20’

Adjustments were made to the table

9. Avoid inclusion of odds ratios in discussion (should mostly feature in results)

10. In references, for online resources please provide access/citation date

The citation date was included

Reviewer #2: 

Major

1. The abstract could be revised. The methods section is lacking substantial information on: the variables, outcome definition, statistical methods used for the analysis. Similarly, there is substantial repeated information in the introduction, methods and results. In the conclusions section, the authors could also include some practical application of the results for targeting policies that reduce these health inequalities in Colombia.

2. Introduction - Rows 73-77. This is an interesting paragraph, but hard to follow and not related to the specific objective of the study. I suggest that this bit is removed to give flow to the introduction.

The phrase was removed

3. Introduction - The authors do not show the readers the current research gaps that this study aim to fulfil.

“Studies on the results of treatment in MDR/RR-TB in Colombia are scarce and located in some cities that are not representative of the Colombian population”

4. Methods - Page 186, it is still not clear if the authors included all the variables with p<0.05 in the bivariate analysis in the multivariate analysis, without testing for interactions, collinearity, or significance. The authors explained their criteria in the discussion, but it would be clearer in the methods section.

“The variables that obtained p values < 0.05 in the bivariate analysis were included in the multivariate analysis.”

5. Methods- In row 220, why there is 67.8% who were affiliated with the subsidized health regime but only 63.7% were treated in “public HCIs”? Public institutions attend both individuals under the subsidized Individuals and the contributory system? If these variables are very correlated (as explained in the discussion), would not be important to choose only one for the multivariate analysis.

In Colombia, the institutions that administer health benefit plans, whether they are of the subsidized or contributory regime, can contract the provision of health services in institutions that provide public or private health services. However, the administrators of the subsidized regime must contract a mandatory percentage with the institutions that provide public health services

6. In the conclusions section, the last paragraph lack of a public health message that interprets or suggests

Minor

7. In the introduction, row 63, when listing the reasons the “and” is missing.

“And” was included

8. In the introduction, Row 69, please replace incidence for either “incident cases”.

“They accounted for only 32% of the estimated incident cases”

9. In row 71, substitute “16,000 estimated cases” by “approximately 16,000 cases”

“Colombia had approximately 16,000 estimated cases of TB”

10. In Table 1, please include the N´s for the columns: “Favorable (N=XX)” and “Unfavorable (N=XX)”

The n was included in each of the groups

11. Table 1, row “Male”, please change the comma for a point. Check commas used instead of points

These corrections were made

12. Figure 1 title, please change “Flow chart” for “flowchart”; Please consider using a title that describes better the figure, e.eg. “Flowchart with selected studied population with MDR/RR-TB in Colombia…”

The title of figure 1 has been corrected

---

## [Decision Letter · Decision Letter 2]

19 Jan 2021

PONE-D-20-18648R2

Factors associated with unfavorable treatment outcomes in patients with rifampicin-resistant tuberculosis in Colombia 2013 – 2015: A retrospective cohort study

PLOS ONE

Dear Dr. Chaves Torres,

Thank you for submitting your revised manuscript to PLOS ONE. After careful consideration, we feel that it has merit but still does not fully meet PLOS ONE’s publication criteria as it currently stands. In particular, while Reviewer #2 is happy with your revision, Reviewer #1 still made comments and suggestions for improvement. Therefore, we invite you to submit a further revised version of the manuscript that addresses the points raised during the review process.

We look forward to receiving your revised manuscript.

Kind regards,

Olivier Neyrolles

Academic Editor

PLOS ONE

Reviewers' comments:

Reviewer's Responses to Questions

**Comments to the Author**

1. If the authors have adequately addressed your comments raised in a previous round of review and you feel that this manuscript is now acceptable for publication, you may indicate that here to bypass the “Comments to the Author” section, enter your conflict of interest statement in the “Confidential to Editor” section, and submit your "Accept" recommendation.

Reviewer #1: (No Response)

Reviewer #2: All comments have been addressed

2. Is the manuscript technically sound, and do the data support the conclusions?

Reviewer #1: Partly

Reviewer #2: Yes

3. Has the statistical analysis been performed appropriately and rigorously? 

Reviewer #1: Yes

Reviewer #2: Yes

4. Have the authors made all data underlying the findings in their manuscript fully available?

Reviewer #1: Yes

Reviewer #2: Yes

5. Is the manuscript presented in an intelligible fashion and written in standard English?

Reviewer #1: No

Reviewer #2: Yes

6. Review Comments to the Author

Reviewer #1: Dear Dr Chaves Torres,

Many thanks for your most recent submission, entitled 'Factors associated with unfavourable treatment outcomes in patients with rifampicin resistant tuberculosis in Colombia 2013 – 2015: A retrospective cohort study' and for the opportunity to review your work.

I have attached more detailed feedback, and have structured my comments using minor and major criteria. The introduction, methodology and conclusion sections would particularly benefit from further review.

I look forward to your revised submission -

Kind regards,

Dr Louisa Chenciner

Reviewer #2: Dear authors, I appreciate the opportunity to revise this interesting piece of work that highlight its importance for TB control policies among the poorest in Colombia and that bring light to the problem of underfinacing of public healthcare systems in Latin America.

7. PLOS authors have the option to publish the peer review history of their article (what does this mean?). If published, this will include your full peer review and any attached files.

Reviewer #1: **Yes: **Louisa Chenciner

Reviewer #2: **Yes: **Julia M Pescarini

---

## [Author Response · Author response to Decision Letter 2]

10 Feb 2021

Response to Reviewers

Ref: PONE-D-20-18648

Factors associated with unfavorable treatment outcomes in patients with rifampicin-resistant tuberculosis in Colombia

Revisions:

Major:

1. Abstract: Background could be made clearer – and consider avoiding use of jargon ‘health benefit plan administrators’ and ‘control entity’ Conclusion – suggest rephrasing ‘such outcomes were associated with subsidized health regime affiliation and age > 60 years’; as previously suggested ‘health regime’ is not immediately clear

Minor:

1. Abstract Methods – specify type of logistic regression model e.g., multivariate Results – aim for consistency in formatting e.g. n =; %; suggest ‘revealed’ is too informal for academic writing

“Background: Multidrug-resistant rifampicin-resistant tuberculosis (MDR/RR-TB) requires prolonged and costly treatment, which is difficult to sustain in the Colombian health system. This requires the joint action of the Health Promoting Companies and health service providers, under the direction, regulation, supervision, monitoring and control of the national government and the Ministry of Health, to provide timely health services to people with TB. The identification of factors associated with unfavorable treatment outcomes in MDR/RR TB patients who received drug therapy between 2013 and 2015 in Colombia can help guide the design of national public health strategies.

Method: A retrospective cohort study was conducted with all patients who received treatment for MDR/RR-TB between January 2013 and December 2015 in Colombia and were registered and followed up by the national TB control program. A multivariate logistic regression model was used to analyze the associations between the exposure variables with the response variable (treatment outcome).

Results: A total of 511 patients with MDR/RR-TB were registered and followed up by the national TB control program in Colombia, of whom 16 (3.1%) had extensive drug resistance, 364 (71.2%) had multidrug resistance, and 131 (25.6%) had RMP monoresistance. The mean age was 39.9 years (95% confidence interval (CI): 38.5-41.3), most patients were male 285 (64.6%), and 299 (67.8%) people qualified for subsidized health care services. The rate of unfavorable treatment outcomes in the RR-TB cohort was 50.1%, with rates of 85.7% for patients with extensive drug resistance, 47.6% for patients with multidrug resistance, and 52.6% for patients with RMP monoresistance. The 511 MDR/RR-TB patients were included in bivariate and multivariate analyses, patients age ≥ 60 years (crude odds ratio (ORc) = 2.4, 95% CI 1.1 – 5.8; adjusted odds ratio (ORa) = 2.7, 95% CI 1.1 – 6.8) and subsidized health regime affiliation (ORc = 3.6, 95% CI 2.3 – 5.6; ORa = 3.4, 95% CI 2.0 – 6.0) were associated with unfavorable treatment outcomes. 

Conclusion: More than 50% of the patients with MDR/RR-TB in Colombia experienced unfavorable treatment outcomes, and the patients who were eligible for subsidized care or were older than 60 years were more likely to experience unfavorable treatment outcomes.”

Major:

2. Introduction Opening paragraph would benefit from restructuring and more focussed approach to the study context 

“Represents a threat to the control of TB because its treatment is more expensive, more toxic and its prognosis is much worse than that of those infected by sensitive strains. Non-adherence to treatment and non-implementation of the directly observed treatment (DOT) strategy have been associated with the development of MDR/RR-TB. In addition other factors, such as sociodemographic barriers that impede access to medicines (e.g., living in rural areas, not having health insurance).”

Sentence 88-90 remains unclear

“Also the proportion of unfavorable treatment outcomes in MDR/RR-TB is higher than that in TB with drug-sensitive bacilli.”

Minor:

Suggest use of ‘adherence’ as opposed to ‘compliance’ 

“Non-adherence to treatment and non-implementation of the directly observed treatment (DOT) strategy have been associated with the development of MDR/RR-TB.”

and ensure consistently used Clarify whether WHO estimates on line 70-73 are globally

“The World Health Organization (WHO) estimated 484,000 cases (range, 417,000-556,000) of MDR/RMP-resistant (RR)-TB globally in 2018 compared to 160,684 cases in 2017”

 Suggest restructuring of sentence 81-85 e.g. ‘Different factors, including education, race, age etc. are associated with higher likelihood of unfavourable outcome’ Suggest use of ‘treatment outcomes’ as opposed to ‘results of treatment’ for consistency 

“Different factors, including education, race, age, drug use, history of second-line treatment, resistance to fluoroquinolones, positive sputum smear after two months of treatment, and XDR-TB are associated with higher likelihood of unfavourable outcome. [4,5,6], and coinfection with HIV has been specifically associated with death[5].”

Are there studies which you could cite here – are they mostly focussed on the capital? 

“Studies on the treatment outcomes results of treatment for MDR/RR-TB in Colombia are scarce and located in some cities that are not representative of the Colombian population”

-line 87 Suggest on line 94 – instead of ‘thus decrease’ consider ‘may contribute to reduction in unfavourable treatment outcome’ – treatment failure is slightly different

“may help guide the design of national public health strategies aimed at improving the care provided to patients with MDR/RR-TB may contribute to reduction in unfavourable treatment outcome”

Major:

3. Methods suggest clearer interpretation of HDI and Gini – as not all readers will immediately understand their significance here 

“Although it has managed to surpass the minimum achievements, placing it in the high human development category (Human Development Index (HDI) of 0.761). There is a poor redistribution of income that perpetuates inequality.”

Are ‘benefit plan administrations’ essentially ‘third party administrators’ – it would be helpful if this terminology is clearer Does the National Public Health plan fall under the remit of the state, the health service providers and ‘benefit plan administrations’ or operate independently – I think this could be explained more clearly and succinctly

“This structure also includes the National Public Health Plan, which is incumbent upon the state”

Minor

3. Methods Suggest avoiding use of term ‘developing countries’

Accepted suggestion the term is eliminated.

Consider rephrasing line 134-135 ‘trusting that most patients would complete treatment’ 

“Data for treatment results were collected until December 2017 assuming a period of at least two years bearing in mind that in these cases the total treatment duration should be at least 20 months”

Suggest specifying this data is anonymised or depersonalised on line 138 

“The Ministry of Health and Social Protection of Colombia (MINSALUD) provided the participants’ data in a database with individualized and anonymised information.”

Reconsider use of language in paragraph line 165-168 – use of term ‘abandonment’ and ‘treatment was finished’ 

“Treatment was favorable when the patient was considered cured or finished treatment. An unfavorable outcome was considered when the result was treatment abandonment, treatment failure, or death.”

Line 188 suggest specifying ‘a logistic regression model’

“A multivariate logistic regression model was used”

4. Results Table 1 – please specify whether these are row or column % in legend and amend table accordingly and ensure use of decimal points only please (see male p value) Please reconsider use of term ‘mestizos’ – does not seem appropriate for use in academic writing

Suggestions were accepted, corrections were made to the table.

Minor

4. Results Ensure consistency in how results are reported (n = X; %)

Suggestions were accepted 

 Suggest rephrasing line 239 ‘significant association with’

“Age ≥ 60 years showed association with unfavorable outcomes”

Major:

5. Discussion Can we be sure that the subsidized health system’s association with unfavourable treatment outcome relates to organisational barriers? Isn’t subsidized healthcare a proxy for social deprivation in your study?

“and we speculate that in addition to the social and economic deprivation to which the population of the subsidized regime may be exposed, there are differences in the care provided to patients between the two health affiliation regimes”

 Out of pocket expenses relating to transport are also considered to be indirect costs – see line 289-291

“In addition, patients incur out-of-pocket expenses, which may be direct, such as those related to transportation and examinations or consultations or indirect, such as an inability to work due to the disease”

Minor:

7. Discussion Avoid use of ‘i.e’ in academic writing if able – see line 275 

i.e eliminated

Consider rephrasing sentence 291-293 

“These expenses in in low-income families can create significant barriers to both access to the TB control program and compliance with TB treatment.”

Consider rephrasing 299 ‘the outcome of drug treatment for TB was favourable for beneficiaries of the BFP’ – implies this was the case for all BFP recipients 

“This relationship between poverty and TB was studied in Brazil, where the authors concluded that TB patients who were beneficiaries of a government cash grant, in this case the Bolsa Familia Program (PBF), obtained favorable results in the TB treatment.”

What do you mean by ‘length of stay in the TB control program’? 

“and the patient's follow-up time in the tuberculosis control program.”

Line 333 – please clarify what is meant by ‘health status’ here? 

“The absence of these variables may lead to overestimation or underestimation of the association between health status and TB treatment outcomes in this cohort of patients.”

Suggest shortening sentence line 335-338 

“The missing data were 70/511, representing a loss of 13.7%, which is acceptable, and the missing cases did not have different characteristics compared to those included in the analysis”

Suggest ‘likely representative’ is used on line 338

“the results obtained are likely representative of the Colombian population”

6. Conclusion Remain unconvinced that you can conclude ‘the structure of the Colombian health care model influences MDR-TB treatment results’ based on your results – given there are several missing confounding variables. Consider revision

“Our results suggest that in addition to the conditions of vulnerability to which the Colombian population affiliated to the subsidized health care system may be exposed, the structure of the Colombian health care model may influence MDR-TB treatment outcomes, which should be considered in the design of public health policies.”

---

## [Decision Letter · Decision Letter 3]

25 Feb 2021

PONE-D-20-18648R3

Factors associated with unfavorable treatment outcomes in patients with rifampicin-resistant tuberculosis in Colombia 2013 – 2015: A retrospective cohort study

PLOS ONE

Dear Dr. Chaves Torres,

Thank you for submitting your revised manuscript to PLOS ONE. Your manuscript will be accepted for publication once you have addressed the remaining comments raised by Reviewer #1, which are only dealing with style and typos. Therefore, we invite you to submit a final revised version of the manuscript that addresses the points raised during the review process.

We look forward to receiving your revised manuscript.

Kind regards,

Olivier Neyrolles

Academic Editor

PLOS ONE

Journal Requirements:

Reviewers' comments:

Reviewer's Responses to Questions

**Comments to the Author**

1. If the authors have adequately addressed your comments raised in a previous round of review and you feel that this manuscript is now acceptable for publication, you may indicate that here to bypass the “Comments to the Author” section, enter your conflict of interest statement in the “Confidential to Editor” section, and submit your "Accept" recommendation.

Reviewer #1: (No Response)

2. Is the manuscript technically sound, and do the data support the conclusions?

Reviewer #1: Yes

3. Has the statistical analysis been performed appropriately and rigorously? 

Reviewer #1: Yes

4. Have the authors made all data underlying the findings in their manuscript fully available?

Reviewer #1: Yes

5. Is the manuscript presented in an intelligible fashion and written in standard English?

Reviewer #1: No

6. Review Comments to the Author

Reviewer #1: Dear Authors,

Many thanks for your revised submission, and for your continued efforts to refine and improve this manuscript.

Please see attached PDF for detailed comments and recommendations to ensure the manuscript is written in standard English, and avoids potentially stigmatising language with respect to TB.

Kind regards,

Dr Louisa Chenciner

7. PLOS authors have the option to publish the peer review history of their article (what does this mean?). If published, this will include your full peer review and any attached files.

Reviewer #1: **Yes: **Louisa Chenciner

---

## [Author Response · Author response to Decision Letter 3]

17 Mar 2021

Revisions:

 Minor Revisions 

Abstract 

Line 28-29: Unclear what ‘Health Promoting Enterprises’ are, and unclear why capitalised. Suggest avoiding using jargon here – and just reference joint action of different service providers 

“This requires the joint action of different providers to provide timely health services to people with TB”

Lines 28-31: Suggest condensing this sentence 

“This requires the joint action of different providers to provide timely health services to people with TB”

Line 34: ‘National public health strategies’ or national TB control programmes? Suggest keeping it specific and focussed 

“Identifying factors associated with unfavorable treatment outcomes in patients with MDR/RR-TB who received drug therapy between 2013 and 2015 in Colombia can help guide the strengthening of the national TB control program.”

Line 37: suggest substituting ‘and were registered’ with ‘who were registered’ 

“who were registered and were registered and followed up by the national TB control program”

Line 38 and Line 193: suggest substituting ‘analyze’ with ‘estimate’ 

“was used to estimate the associations between the exposure variables with the response variable (treatment outcome)”

Line 44: suggest omitting ‘people’ in ‘299 (67.8%) people qualified’ 

“were eligible for subsidized health services”

Line 54 -55: suggest separating into two separate sentences for clarity e.g. ‘..unfavourable treatment outcomes [FULL STOP] Patients who were eligible for subsidized care or over 60 years of age were more likely to experience unfavourable treatment outcomes.’ 

“The patients who were eligible for subsidized care were more likely to experience unfavorable treatment outcomes. Those who were older than 60 years were also more likely to experience unfavorable treatment outcomes”

Introduction

Line 68: Please revise this sentence – suggest ‘MDR/RR-TB poses a threat to TB control – patients have worse (long term?) prognosis, and treatment is more likely to be expensive and present toxicity’ 

Suggestion accepted, it was corrected in document

Line 70-71: Suggest ‘lack of treatment support (including DOT) and limited treatment adherence have both been associated with the development of MDR/RR-TB’ 

Suggestion accepted, it was corrected in document

Line 72-73: Ensure this sentence makes grammatical sense – suggest ‘Other contributing factors include social and cultural barriers to access, such as living in rural areas and the absence of (private?) health insurance’ 

Suggestion accepted, it was corrected in document

Line 82-84: ‘Also’ not necessary here – Consider ‘The proportion of patients experiencing unfavourable treatment outcome is higher in MDR/RR-TB than in TB with drug sensitive bacilli’ 

Suggestion accepted, it was corrected in document

Line 84: Suggest ‘Other factors’ not ‘Different factors’ 

Suggestion accepted, it was corrected in document

Line 90: Consider cutting ‘additionally’ and cutting ‘requires the joint action of different actors within the same system to provide health services’ instead ‘MDR/RR-TB often necessitates prolonged and more expensive treatment, which is challenging to sustain in the current Colombian health care system’ 

Suggestion accepted, it was corrected in document

Line 93-97: Suggest condensing to ‘Identifying factors associated with unfavourable treatment outcome for patients with MDR/RR-TB in Colombia, may support design and delivery of the national TB control programmes’ 

Suggestion accepted, it was corrected in document

Line 99: Suggest ‘treatment’ not ‘pharmacotherapy’ for consistency please 

Suggestion accepted, it was corrected in document

Methods 

Line 103: Please omit ‘it has managed to surpass the minimum achievements’ – suggest ‘Colombia is an upper-middle income country (Human Development Index 0.761), however there remains significant wealth inequality’

Suggestion accepted, it was corrected in document

Line 106: Suggest omitting ‘the social security system for health care’ and instead ‘the public health system encounters…’ 

Suggestion accepted, it was corrected in document

Line 109-112: Suggest separating into two sentences e.g. ‘…95% of the population [STOP]. This organization separates insurance and administration of financial resources, from the (direct?) service provision management of its members’ – still not fully clear what service provision management entails 

Suggestion accepted, it was corrected in document

Line 117-118: Consider ‘who rely on public healthcare services’ instead of ‘for whom health services are covered by government resources’ 

Suggestion accepted, it was corrected in document

Line 118: use ‘qualifying’ not ‘qualified’ 

Suggestion accepted, it was corrected in document

Line 121: apologies – still unclear what ‘Health Promoting Enterprises’ are – and I wonder if you are referencing ‘Health Solidarity Enterprises’ (Empresas Solidarias de Salud, ESSs) instead? https://jech.bmj.com/content/jech/56/10/742.full.pdf - 

Are not "solidarity health companies". Health insurance companies are private health insurance companies. https://doi.org/10.1080/09581596.2018.1449943

Please clarify Line 124: Please condense ‘including healthcare institutions, such as..’ to ‘including hospitals and community clinics’ 

Suggestion accepted, it was corrected in document

Line 126: Suggest condensing ‘The Colombian Government’s National Public Health Plan prioritises national health promotion and disease prevention, including infectious diseases such as TB, Leprosy and Malaria. Therefore, TB is a disease of public health interest, and its diagnosis, treatment and follow-up are publicly funded as a result’ 

Suggestion accepted, it was corrected in document

Line 135: Grammar – replace ‘received’ with ‘receiving’ 

Ok

Line 136: Suggest ‘treatment outcomes’ in lieu of ‘treatment results’ for consistency 

Ok

Line 136-138: Grammar – more than one clause in this sentence – consider revision – e.g., appropriate punctuation 

“Data for treatment outcomes were collected until December 2017, assuming a period of at least two years bearing in mind that in these cases the total treatment duration should be at least 20 months”

Line 140: Suggest revision of ‘their information was catalogued as missing data’ – if they were excluded then why were they categorised as missing? 

Line 141-142: ‘individualised and anonymised information’ – please clarify which it is – this is contradictory! 

“Patients were excluded if no record of treatment results up to this date was available, and their information was catalogued as missing data. The Ministry of Health and Social Protection of Colombia (MINSALUD) provided the participants’ data in a database with individualized and anonymised information”

Line 169-170: ‘favourable treatment or unfavourable outcome’ – for purposes of consistency please either refer to treatment outcome. Ok

Line 172 and line 183: As previously mentioned – please avoid stigmatising language – ‘treatment abandonment’ 

Was changed to “lost to follow up”

Line 190-191: Again, not entirely clear were these patients excluded through the study inclusion criteria in which case no need to mention again here 

the phrase “Seventy patients categorized as not evaluated were not included in the analysis because the treatment outcome was unknown” was removed from the document

Results 

Line 217: Again, shouldn’t the 70 (13.7%) be included in those who did not meet your inclusion criteria? 

the phrase “and 70 (13.7%) patients were excluded because their treatment outcomes were unknown” was removed from the document

Line 230: Please format (n = ; %) 

“most patients were male (n=285; 64.6%)”

Line 233: Suggest omitting ‘nine of which were capitals’ 

Suggestion accepted, it was corrected in document

Line 238-239: Suggest two separate sentences e.g. ‘…RR-TB showed unfavourable treatment outcomes. [STOP] 122 (27.7%)’ 

Suggestion accepted, it was corrected in document

Discussion 

Line 268-269: Suggest two separate sentences and condense ‘was found in this MDR/RR-TB cohort. [STOP]’ 47.4% of patients with monoresistance to RMP experienced favourable outcome…’ 

“An overall therapeutic success rate of 49.9% was found in this MDR/RR-TB cohort. 47.4% among patients with monoresistance to RMP, 52.4% among patients with MDR-TB, and 18.6% among patients with XDR-TB.”

Line 279: Again, please keep language consistent TB treatment outcomes, not ‘results of TB treatment’ 

“The association between the health regime and the TB treatment outcomes”

Line 280: Suggest two separate sentences ‘is relevant. [STOP] Individuals affiliated with…’ 

Suggestion accepted, it was corrected in document

Line 283-289: This sentence needs to be split into two/three – important discussion, but can be made clearer through restructuring. 

“Although the diagnosis and treatment of MDR/RR-TB in Colombia are free for all patients regardless of their health regime affiliation, discrepancies remain in the outcomes of treatment. We speculate that in addition to the social and economic deprivation to which the population of the subsidized regime may be exposed, there are differences in the care provided to patients between the two health affiliation regimes. Which can be considered an organizational barrier, a factor hindering initial contact with health services (entry barriers) and timely care after a patient enters the health center (interior barriers)”

Line 290: suggest ‘a national level’ not ‘the national level’ 

Suggestion accepted, it was corrected in document

Line 290-294 suggest condensing ‘for patients receiving care in the subsidised healthcare system, higher under-5 mortality, maternal mortality and mortality related to communicable disease has already been observed’ 

Suggestion accepted, it was corrected in document

Line 298 Please avoid stigmatising language – suggest ‘adherence’ not ‘compliance’

Suggestion accepted, it was corrected in document

Line 298-300: I think you could relate this more closely to your results to strengthen your argument! 

“These expenses in in low-income families (As is the case for families in the subsidized regime) can create significant barriers to both access to the TB control program and adherence with TB treatment”

Line 305-306: suggest ‘were more likely to experience favourable outcome’ not ‘obtained favourable results’ 

Suggestion accepted, it was corrected in document

Line 331-332: suggest omitting ‘thus all available variables were analyzed’ 

Suggestion accepted, it was corrected in document

Line 332: avoid use of ‘influence’ – this implies extent of temporal relationship 

Line 333: what do you mean by ‘exclude confounding factors’ – I’m not sure your analysis is able to achieve this 

Suggestion accepted, it was corrected in document

Line 338: please be more specific about the nature of bias this may introduce and the effect on your main findings 

“Due to a selection bias, which prevents extrapolating our results to the Colombian population with any of the aforementioned conditions”

Line 340: please avoid use of fractions interspersed in text e.g. ‘70/511’ 

Suggestion accepted, it was corrected in document

Line 346: what do you mean by ‘terms of updating’ consider rephrasing 

“it can still be improved the quality of the recorded data to reduce lost data and facilitate collection of the largest number of variables possible in reports for researchers who request these data”

Line 355-356: Suggest keeping conclusion more specific e.g. design of TB control programs, or strategies to reduce burden of MDR/RR-TB

---

## [Editor Report · Decision Letter 4]

22 Mar 2021

Factors associated with unfavorable treatment outcomes in patients with rifampicin-resistant tuberculosis in Colombia 2013 – 2015: A retrospective cohort study

PONE-D-20-18648R4

Dear Dr. Chaves Torres,

We’re pleased to inform you that your manuscript has been judged scientifically suitable for publication and will be formally accepted for publication once it meets all outstanding technical requirements.

Kind regards,

Olivier Neyrolles

Section Editor

PLOS ONE

---

## [Editor Report · Acceptance letter]

5 Apr 2021

PONE-D-20-18648R4 

Factors associated with unfavorable treatment outcomes in patients with rifampicin-resistant tuberculosis in Colombia 2013 – 2015: A retrospective cohort study 

Dear Dr. Chaves-Torres:

I'm pleased to inform you that your manuscript has been deemed suitable for publication in PLOS ONE. Congratulations! Your manuscript is now with our production department. 

Kind regards, 

on behalf of

Dr. Olivier Neyrolles 

Section Editor

PLOS ONE